



# Seismic monitoring of the STIMTEC hydraulic stimulation experiment in anisotropic metamorphic gneiss

Carolin M. Boese[1], Grzegorz Kwiatek[1], Thomas Fischer[2], Katrin Plenkers[3], Juliane Starke[1], Felix

Blümle[1,4], Christoph Janssen[1],Georg Dresen[1]

[1]Helmholtz Centre Potsdam, GFZ German Research Centre for Geosciences, Section 4.2: Geomechanics and
Scientific Drilling, Telegrafenberg, 14473 Potsdam, Germany
[2]GmuG mbh, Bad Nauheim, 61231, Germany
[3]ETH Zurich, Bedretto Lab, NO F27, 8092 Zürich
[4]ASIR Seismic GmbH, Aachen, 52062, Germany

*Correspondence to*: C. M. Boese (carolin.boese@gfz-potsdam.de)

**Abstract.** In 2018 and 2019, the STIMTEC hydraulic stimulation experiment was conducted at 130 m depth in the
Reiche Zeche underground research laboratory in Freiberg/Germany. The experiment was designed to investigate the
rock damage resulting from hydraulic stimulation and to link seismic activity and enhancement of hydraulic
properties in strongly foliated metamorphic gneiss. We present results from active and passive seismic monitoring
prior to and during hydraulic stimulations. We characterise the structural anisotropy and heterogeneity of the
reservoir rocks at the STIMTEC site and the induced, high-frequency (>1 kHz) acoustic emission (AE) activity,
associated with brittle deformation at the cm to dm-scale. We derived the best velocity model per recording station
from over 200 active ultrasonic transmission measurements for high accuracy AE event location. The average P-
wave anisotropy is 12%, in agreement with values derived from laboratory tests on core material. We use a 16-
station, seismic monitoring network comprising AE sensors, accelerometers, one broadband sensor and one AE-
hydrophone. All instrumentation was removable, providing us with the flexibility to use existing boreholes for
multiple purposes. This approach also allowed for optimising the (near) real-time passive monitoring system during
the experiment. To locate AE events, we tested the effect of different velocity models and inferred their location
accuracy. Based on the known active ultrasonic transmission measurement points, we obtained an average relocation
error of 0.26±0.06 m using a transverse isotropic velocity model per station. The uncertainty resulting from using a
simplified velocity model increased to 0.5–2.6 m, depending on whether anisotropy was considered or not. Structural
heterogeneity overprints anisotropy of the host rock and has a significant influence on velocity and attenuation, with
up to 4% and up to 50% decrease on velocity and wave amplitude, respectively. Significant variations in seismic
responses to stimulation were observed ranging from abundant AE events (several thousand per stimulated interval)
to no activity with breakdown pressure values ranging between 6.4 and 15.6 MPa. Low-frequency seismic signals
with varying amplitudes were observed for all stimulated intervals that correspond to the injection pressure curve
rather than the flow rate. We discuss the observations from STIMTEC in context of similar experiments performed
in underground research facilities to highlight the effect of small-scale rock, stress and structural heterogeneity
and/or anisotropy observed at the decameter scale. The reservoir complexity at this scale supports our conclusion that
field-scale experiments benefit from high-sensitivity, wide-bandwidth instrumentation, and flexible monitoring
approaches to adapt to unexpected challenges during all stages of the experiment.



## 1 Introduction

Meso-scale, in-situ hydraulic stimulation experiments performed in well-instrumented underground research laboratories (URL) offer a number of advantages over small-scale laboratory tests and reservoir-scale experiments. In particular, URL experiments capture structural heterogeneity on a realistic length scale and are thus essential to transfer results from laboratory tests on centimetre-scale rock samples to reservoir rocks at the kilometre-scale (Young et al. 2000; Gischig et al., 2019). Furthermore, URL experiments allow for validation of inferred results, e.g., through mine-back drilling into stimulated rock volumes (e.g., Warren and Smith, 1985). Most importantly, intermediate-scale, in-situ experiments, conducted in URLs, allow for a close to optimal placement of seismic sensor networks for monitoring and characterisation of the target volume (Ohtsu, 1991; Zang et al. 2017; Amann et al., 2018; Kwiatek et al., 2018; De Barros et al., 2019; Feng et al. 2019). Hydraulic stimulation was seismically monitored during in-situ experiments in various settings (e.g. Ohtsu, 1991; Dahm et al. 1999). The monitoring systems need to be tuned to the seismic waves associated with hydraulic stimulation in terms of sensitivity, frequency range and attenuation characteristics of the reservoir, which limit the detection ranges of the seismic signals (e.g. Mendecki et al., 1999; Plenkers et al. 2010, 2011; Manthei and Plenkers, 2018). Varying noise conditions on site often impact monitoring conditions (Plenkers et al., 2010, 2013). Recently, monitoring of a hydraulic stimulation experiment at 410 m depth at the Äspö Hard Rock Laboratory (AHRL) in southern Sweden in May/June 2015 (Zang et al., 2017; Kwiatek et al., 2018) showed that only two of the multiple seismic monitoring systems in place were suitable to record the observed seismic processes. The high-sensitivity acoustic emission (AE) network recorded high-frequency (>1 kHz) AE events from fracturing and frictional sliding with rupture dimensions on the centimetre to decimetre scale. A five-station broadband network recorded low-frequency signals of 0.004–0.008 Hz during the frac and refracs. Slow deformation processes have also been monitored with tilt sensors during the "In-situ Stimulation and Circulation Experiment" performed at Grimsel Test Site (GTS) in Switzerland. This experiment was conducted at a depth of 480 m below surface, within an experimental volume of ca. 20 m × 20 m × 20 m of granitic rock between February and May 2017 (Gischig et al., 2018). Dense 3-D coverage and the close proximity of seismic instrumentation to induced AE events both at the AHRL and the GTS sites resulted in high-quality data sets resolving details of the hydro-mechanical processes on the decimetre to metre scale (e.g., Dutler et al., 2019; Kwiatek et al., 2018; Villiger et al., 2020). This level of detail is necessary to advance our understanding of processes relevant for hydraulic stimulations such as (1) hydro-mechanically coupled fluid flow and pore pressure propagation, (2) transient pressure-dependent and permanent slip-dependent permeability changes, (3) fracture formation and interaction with pre-existing structures, (4) rock mass deformation around the stimulated volume due to fault slip, failure processes and poroelastic effects, and (5) the transition from aseismic to seismic slip (e.g. Amann et al., 2018). Currently, AE event distributions provide the most detailed information on small-scale spatio-temporal-evolution of the deformation within the reservoir induced by hydraulic stimulation. In particular, fracture dimensions, orientations, faulting style, and the orientation of the prevailing principal stress axes may be inferred from the analysis of induced seismic events (Manthei et al 2001; van der Baan et al., 2013; Manthei and Plenkers, 2018; Krietsch et al., 2019).

The STIMTEC experiment was designed to develop diagnostic criteria for successful hydraulic stimulations, and to optimise monitoring and stimulation procedures. This experiment was conducted in strongly foliated and



heterogeneous metamorphic rock at shallow depth (~130 m). Complementary to the STIMTEC experiment, several
other meso-scale injection experiments in crystalline rock are currently underway. The "EGS Collab Experiment" is
a multi-institutional collaborative research project at a similar scale that aims to solve technological problems related
to EGS-reservoir creation and operation through different stimulation procedures under realistic in situ stress
conditions, and to provide a test bed for the validation of existing thermal-hydrological-mechanical-chemical
numerical modelling tools (Kneafsey et al., 2018). The second experimental phase is currently planned at the Sanford
Underground Research Facility (SURF) at 1.25 km below surface, located in the Homestake mine, a former gold
mine in South Dakota, USA (Kneafsey and the EGS Collab Team, 2020; Schoenball et al., 2020). The Bedretto
experiment aims at upscaling previous meso-scale experiments by a factor of ten (Gischig et al., 2019) and is located
in the Bedretto Underground Laboratory for Geoenergy research (BULG) in Southern Switzerland, about 10 km
southeast of the GTS. Current activities aim at stimulating the Rotondo granite at the Bedretto tunnel with an
overburden about 1 km thick in an estimated volume of ca. 300 m × 100 m × 50 m allowing to test different
hydraulic stimulation as well as seismic and deformation monitoring techniques.

Site complexity due to small-scale rock stress and structural heterogeneity and/or anisotropy of varying strength and
orientation is a major issue encountered by all meso-scale in situ experiments so far. To trace the spatio-temporal
evolution of AE events during hydraulic stimulations at high resolution, the accuracy of the applied seismic velocity
model for location in anisotropic and heterogeneous rock volumes is of fundamental importance. At the laboratory
scale, anisotropic velocity models are commonly applied (e.g., Stanchits et al., 2003). The models are fundamentally
important to monitor rock-deformation during laboratory tests at high resolution. At the mine scale, comprehensive
and dense in-situ measurements, in particular active seismic surveys, are performed to characterise heterogeneity and
anisotropy of the investigated rock volume. These seismic surveys are commonly performed before the stimulation
to derive the velocity structure and repeatedly in material science and in-situ experiments to monitor alteration of the
rock volume e.g. by fracture generation. Repeated active measurements throughout hydraulic stimulation experiment
are still scarce. Their value for monitoring temporal changes resulting from fluid pressure changes in the rock
volume has only recently been recognized (Doetsch et al., 2018; Rivet et al., 2016; Schopper et al., 2020). At the
field scale, detailed site characterisation is often not possible because of associated costs and limited placement of
instrumentation, resulting in velocity model ambiguity and lower resolution of the seismic event distribution. Thus,
in STIMTEC we performed resolution tests at the meso-scale to place better constraints on model uncertainties and
to provide estimates of the effect of simplifications and approximations required at the field scale.

The seismic response to stimulation during recent URL experiments was highly variable. At the AHRL site seismic
response to stimulation likely depended on rock-type with granodiorite and granite stimulations showing seismicity
in contrast to diorite-gabbro host rocks. However, this interpretation is complicated by the fact that three different
fluid-injection schemes were applied to test their influence on injectivity and induced seismicity (Zang et al., 2013).
At the GTS site, two shear zones (S1, S3) with different deformation history in the Grimsel granodiorite were
stimulated. Hydrofrac experiments revealed remarkably different seismic responses north and south of the S3 shear
zone in terms of injection pressure, amount of backflow, injectivity before jacking and final transmissivity (see Fig. 4
and 5 of Dutler et al., 2019). Villiger et al. (2020) observed differences in the seismicity patterns observed during



hydroshear stimulation of the two shear zones. During stimulation of the S1 shear zones, the majority of AE events
occurred at the beginning of injection, when the total volume of injected fluid was low, whereas for the S3 shear
zone the number of AE events increased with the volume of injected fluid (Villiger et al., 2020). Hydroshear
stimulations of the ductile S1 shear zone showed less seismicity overall and larger transmissivity increases than S3
hydroshear stimulations. The seismic responses to stimulation during the EGS Collab experiment were also complex
(Schoenball et al., 2020). Abundant seismicity accompanied the three hydraulic stimulations at 1.5 km depth at
SURF aiming to establish a connection between injection and production boreholes approximately 10 m apart
(Kneafsey et al., 2019). Seismicity delineated at least ten planar features with variable orientations that connected to
an open natural fracture, which formed a significant fluid pathway and controlled the stimulations (Schoenball et al.,
124     2020).

Here, we introduce the STIMTEC project, its monitoring concept and lessons learned from using a 16-station seismic
monitoring network for active and passive seismic monitoring during a decimetre-scale hydraulic stimulation
experiment in anisotropic and heterogeneous rock. We compare our monitoring experience with other previous and
ongoing research experiments in URLs. We review our seismic monitoring strategy, monitoring system adjustments
and discuss potential applications to the field scale. We address how anisotropy and heterogeneity are characterised
and provide estimates to place better constraints on the effect resulting from simplifications and approximations
commonly applied at the field scale.

## 2    The STIMTEC project
### 2.1    Objectives, experimental framework, and monitoring strategy

The STIMTEC experiment focusses on the development and optimisation of hydraulic stimulation (STIMulation
TEChnologies) and aims at establishing the link between damage patterns, hydraulic properties, and observed
seismic activity to provide diagnostic criteria for the success of a stimulation (Renner and STIMTEC team, 2021).
Therefore, seismic and hydraulic monitoring are key components of the experiment. In addition, validation through
mine-back drilling into stimulated volumes of complex rock, small-scale laboratory tests to characterise mechanical
and physical properties and numerical modelling are part of the integrated project approach.
The STIMTEC experiment comprised the following phases:

- a *pre-stimulation characterisation phase* (including site characterization, borehole drilling and logging, core analysis and hydraulic measurements for interval selection, as well as instrumentation);
- *the stimulation phase* (stimulation of ten selected intervals in the injection borehole during 16–18 July 2018);
- *the hydraulic testing phase* (testing of six intervals in the injection borehole during 8–10 August 2018);
- *the validation phase* (mine-back drilling of three validation boreholes, stress measurements in five intervals of the vertical validation borehole on 21/22 August 2019); and
- *the final hydraulic testing phase* (testing of seven intervals in the injection borehole during 5–8 November 2019).



High-resolution seismic monitoring accompanied all experimental phases, but with different foci. During the pre-
stimulation characterisation phase, active seismic monitoring aimed at identifying high-attenuation and deformation
zones to avoid sensor installation in these zones, to quantify detection ranges, and to obtain a velocity model. The
installed sensors were then used to characterise background noise levels and any natural seismicity at the site. During
the stimulation phase and subsequent validation phase, real-time passive monitoring aimed at optimised AE event
detection, localisation and magnitude estimation during stimulation of intervals in the injection and vertical
validation boreholes. Repetitive active seismic measurements were performed along the injection and validation
boreholes to investigate any elastic velocity changes resulting from the stimulation. During the final hydraulic testing
phase, passive seismic monitoring focused on verifying detection rates observed for some stimulated intervals with
few AE event by placing two sensors closer to these intervals.

**2.2   Site description and infrastructure**
The STIMTEC site is located on the second floor of the Reiche Zeche Mine, in the Eastern Ore Mountains beneath
the city of Freiberg, Germany at a depth of ca. 130 m below surface (Figure 1). The metamorphic gneiss complex,
penetrated by the mine, is referred to as the Freiberger gneiss anticline, and belongs to the Precambrian metamorphic
basement of the internal Mid-European Variscan orogeny (Seifert and Sandmann, 2006). It hosts silver, lead and zinc
ores, which were mined for centuries (Bayer, 1999). Temperatures at the STIMTEC site are low (~10℃). The
protolith of the Inner Grey Gneiss at Freiberg likely was an S-type granite (Tichomirowa et al., 2001, and references
therein), which was metamorphosed at about 0.8 to 1.1 GPa and 600 to 700°C and has a Proterozoic age with
minimum estimates of 548 to 534 Myrs (c.f. Fig. 11 of Tichomirowa et al., 2001). The fine-grained biotite gneiss has
a granitic appearance and often contains large potassium-feldspar porphyroblasts. The mineral composition of
Freiberg gneiss is generally characterized by biotite, potassium feldspar, plagioclase, and quartz (Tichomirowa et al.,
2001). Freiberg gneiss is a partly weathered, faulted and strongly foliated rock. Large, steeply-dipping mineralized
fault zones strike through the gneiss (Sebastian, 2013).

The monitored rock volume at the STIMTEC site has dimensions of 40 m × 50 m × 30 m and is situated between
two galleries: the straight driftway and the curved vein drift that tracks the mined ore lode "Wilhelm Stehender"
(Figure 1), a major mineralized fault zone with a thickness of up to 2 m that strikes north and dips westward beneath
the site. Large ore lodes at Reiche Zeche are generally considered normal faults and trend predominantly north-south
to northeast-southwest. The galleries have a square cross section (width/height of ca. 2 m) and were excavated in
1903 (vein drift) and 1950 (driftway).

In total, seventeen boreholes with uniform radius (76 mm) were drilled in two phases. Eleven seismic monitoring
boreholes were completed with a range of orientations and lengths, extending horizontally or upwards from the
galleries (Figure 1). The 63 m-long injection borehole was drilled at the maximum inclination angle to the sub-
horizontal foliation compatible with seismic monitoring requirements (upwards directed boreholes, possible
recording ranges, and placement outside of damage zones). It strikes N31°E and dips 15°. A more steeply inclined
(dipping 36°, striking N66°E) hydraulic monitoring borehole was drilled, extending below the central part of the
injection borehole with a minimum distance of 2.5 m between the borehole depth 18.4 m in the hydraulic monitoring



borehole and 33.9 m in the injection borehole. One cable borehole, connecting the two galleries, was drilled for cable as well as seismic sensor installation. The validation phase comprised mine-back drilling of two inclined validation boreholes of 19.25 m and 45.8 m length, running sub-parallel to the injection borehole and targeting seismically active and inactive volumes (Figure 1), as well as a vertical borehole for evaluation of the stress field. The short and long inclined validation boreholes dip ~12° and ~15°, terminating 3.5 m above and 4.4 m sideways of borehole depth 28.1 m and 56.6 m in the injection borehole, respectively. The 15.6m-long vertical validation borehole (dip angle of ~89°) is located in the driftway and spans the same absolute depth range as the injection borehole.

The STIMTEC site is located 180 m south of the GFZ underground laboratory (Giese and Jaksch, 2016), where extensive site investigations and exploration monitoring in the 10–3000 Hz frequency range have been performed over the last 20 years to characterize the rock mass. The excavation damage zone (EDZ) of the galleries at the GFZ lab may extend up to 10 m into the rock volume with an estimated 7% reduction in P-wave velocity (Krauß et al., 2014). A continuous east-west trending damage zone was seismically imaged showing a ca. 13% P-wave velocity reduction compared to the surrounding rock mass (Krauß et al., 2014). Predominantly east-west trending structures are likely relicts given their orientation with respect to the current regional stress field. The stress field was measured at 140 m depth in the mine, a few hundred metres from the STIMTEC site using an overcoring technique (Table 1; Mjakischew, 1987), suggesting a strike-slip regime with maximum horizontal compressive stress oriented NNW-SSE, which is typical for SE Germany.

## 2.3    Structural analyses

Geological structures within the STIMTEC rock volume were identified through mapping of the access galleries, acoustic televiewer images of the injection, hydraulic monitoring and validation boreholes, and from inspection of the recovered core material. This aimed at the detection of possibly continuous fracture systems or damage zones, which could affect the recording of high-frequency acoustic emission events. The foliation was mapped at 34 positions and determined to be sub-horizontal to shallowly dipping in a south-east-direction. At least two, east-west trending, steeply-dipping deformation zones were identified in both galleries that occasionally serve as water conduits as indicated by oxidation and $Fe_2O_3$ deposition in the otherwise intact rock mass. These are referred to as the northern and southern deformation zone. A third zone, the 'middle deformation zone', was predominantly seen in vein drift. Drilling and coring of the injection and validation wells allowed us to check whether these deformation zones actually crossed the entire STIMTEC volume (question mark in Figure 1). The density of open fractures identified from acoustic logs is highest (with 20 fractures per meter) at the bottom of the injection and long inclined validation boreholes, compared to typical values of five open fractures per metre elsewhere. Several prominent structures (at 60 and 62 m) with a range of orientations were identified in the logs from the injection borehole (Figure 2), where the core becomes severely fractured and was not fully recovered. This zone is considered the continuation of the northern deformation zone at depth within the rock volume. Its location and depth is consistent with the orientation of mapped structures in both galleries (Figure 1).

A connection of the middle damage zone between the driftway and the vein drift is not well constrained. A prominent single fracture is mapped at 32.5 m depth in the injection borehole, also seen at 17 m in the hydraulic



monitoring borehole and at 19.8 m in the long, inclined validation borehole (Figure 2). However, this notable
structure was not observed in the short, inclined validation borehole. Its interpreted orientation does not match the
interpolated position of the middle damage zone based on mapping in the galleries. Ultrasonic transmission
measurements from the cable borehole, connecting the two tunnels, indicate that the mapped deformation zone seen
in vein drift extends several meters into the rock volume but does not connect to the driftway.

Between 33–41 m depth in the injection borehole, the number of healed fractures identified from the core is largest.
Two prominent structures are seen at 46 and 47 m depth, located in a section of the injection borehole (42–50 m) that
contains more fractures on average (Figure 2). The same two structures are likely seen at 38–39 m depth in the long
validation borehole.

Based on the distribution of fractures obtained from core analyses and acoustic image logs as well as hydraulic pre-
characterisation results, ten stimulation intervals of 0.75 m length each were selected for stimulation in the injection
borehole. Intact intervals were located at borehole depths of 22.4, 24.6, 28.1, 33.9 and 37.6 m (depths reference to
the position of the middle of the double-packer probe), while intervals with pre-existing fractures were located at
40.6, 49.7, 51.6, 55.7 and 56.5 m depth (Table 2). Four intact sections and one test interval with a pre-existing
fracture were selected for stimulation in the vertical validation borehole, corresponding to 4.0, 6.7, 9.3, 11.7 and 13.2
m depth (Figure 1).

**2.4    Hydraulic injection scheme**
All selected intervals in the injection and vertical validation borehole were stimulated with a uniform fluid injection
scheme:
First, a pulse test was performed in the packed-off interval. The test interval was pressurized to assess the
performance of the packers and to assess the presence or absence of pre-existing open, conductive fractures.
Hydraulic properties were obtained from the time that it takes the pressure to decay from the initial pressure to a
certain level (Bredehoeft and Papadopulos, 1980; Cooper et al., 1967). Secondly, fluid was injected into the packed-
off interval, maintaining a constant flow-rate and thereby raising the interval pressure until breakdown to create a
hydro frac. Once the breakdown pressure was reached the injection was shut-in. Thirdly, three refracs were
performed at the same flow-rate as applied during the frac to determine fracture re-opening pressures, to propagate
the fracture, and to monitor the evolution of shut-in pressures after each refrac. Subsequently, a step-rate test was
performed, comprising stepwise increases of the injected fluid to determine the jacking pressure, when the created
fractures changed their state from mechanically closed to mechanically opened. Optionally, a periodic pumping test
sequence was performed to derive hydraulic properties, consisting of phases of alternating flow-rates between two
levels, ranging from 0.6/1.5 l/min to 6.5/8.5 l/min, for periods varying between 20 s and 900 s (~15 minutes; Table

265    2).


**2.5    Seismic monitoring network and data acquisition**
The seismic monitoring network consisted of 16 sensors, installed in boreholes of 1.5 m to 20 m length to reach
beyond the tunnel excavation damage zone. This sensor network was used for both active seismic measurements and



passive seismic monitoring. We used 12 *GMuG[1] MA BLw-7-70-75* AE side-view single-component in-situ AE
sensors that provided high sensitivity in the frequency range 1–100 kHz, allowing to detect AE events with rupture
plane dimensions in the cm- to dm-scale (cf. Kwiatek et al., 2011; 2018). The AE sensors were placed in upwards
pointing boreholes located above the injection well, reducing the risk of sensor failure due to water intrusion.
Minimum sensor distances to the stimulation intervals in the injection borehole were 5.3–19.7 m (Figure 1, c.f. Table
2 for average distances). The spatial coverage of the sensors was optimised for event detection, determination of
hypocentres and focal mechanisms (cf. Plenkers et al., 2010; Kwiatek and Ben-Zion, 2016), based on results
obtained from an active seismic survey performed in the pre-stimulation characterisation phase. This survey showed
a strong influence of deformation zones on the amplitude and frequency content recorded by the AE sensors and
placed constraints on maximum recording distances. Given the limitations regarding the number of monitoring
stations and expected strong damping of elastic waves, we realised that not all parts of the injection borehole could
be equally well monitored. We therefore focussed the seismic monitoring on the intermediate-depth range (25–35 m
depth) of the injection borehole. However, we decided to drill two monitoring boreholes longer than required for the
preferred network design to allow for changes in sensor placements, if necessary. In addition, one channel of the
datalogger was left available for flexible use and testing onsite.

Three AE sensors were co-located with uniaxial *Wilcoxon 736T* accelerometers with sensitivity between 0.05–25
kHz for the in-situ calibration of the AE sensors (cf. Plenkers et al., 2010; Kwiatek et al., 2011, 2018). In addition, a
six-component *ASIR[2] A-SiA-ULN-G4.5-GS-70* broadband sensor was installed in a borehole to extend the range of
recorded signals to low frequencies. It consists of a three-component 4.5 Hz geophone and a three-component ultra-
low noise optical accelerometer with sensitivity in the range 0.01–100 Hz. This borehole sensor is noisier in the
frequency band 0.01–10 Hz but less noisy for 10–100 Hz compared to the Trillium Compact 120 s broadband
sensors installed in the AHRL tunnels, which recorded low-frequency signals associated with the frac and refracs
(Zang et al., 2017). One component of the sensor was simultaneously recorded on the high frequency AE system data
logger (using the one channel available for flexible use during pre-stimulation and stimulation phases) and by a low-
frequency six-channel broadband system data logger (during all experimental phases) for synchronous timing and
data matching. The broadband sensor was first installed in a 1.5 m long sub-horizontal borehole in the vein drift, but
was then removed and modified for installation in the 15 m- deep vertical validation borehole in the driftway. By
placing the sensor closer and at a comparable absolute depth to the deepest stimulation intervals in the injection
borehole, we wanted to test if it recorded signals associated with stimulation and hydraulic testing of these intervals.

A *GMuG HAE40k* sensor, hereinafter referred to as an AE-hydrophone, because of its characteristics somewhat
similar to an hydrophone and suitable for in-water installation, was installed in the down-going hydraulic monitoring
borehole for the final hydraulic testing phase, and connected to the available channel for flexible use. This
piezoelectric AE-sensor is sensitive to pressure changes in the frequency range 1–40 kHz and was added to the

---

[1]Gesellschaft für Materialprüfung und Geophysik (www.gmugmbh.de)

[2]Advanced Seismic Instumentation and Research LLC (www.asirseismic.com)



network to provide a high-sensitivity sensor in close proximity (6–17 m) to the intermediate and deep stimulation
intervals.

Seismic waveforms were recorded with the *GMuG AE System* datalogger, a 16-channel, 16-bit acquisition system
that allowed recording both in trigger-mode with a sampling frequency of 1 MHz as well as in continuous mode with
sampling frequency of either 200 or 500 kHz. The six channel data logger of the broadband system recorded
continuously at 125 Hz during the initial stimulation and hydraulic testing and 1000 Hz during the final hydraulic
tests. By using a continuous and a triggered seismic monitoring system simultaneously, data redundancy and
different data accuracy was obtained. The two seismic monitoring modes can be easily switched from one to the
other, allowing for flexible use for active (up to 32 channels, in triggered mode) and passive seismic monitoring (16
channels, both modes).

**2.6    Active seismic measurements**
For active measurements three different sources, capable of generating high-frequency signals in the kHz range, were
used. A survey, comprising sledge-hammer hits at 84 fixed positions in the vein drift recorded by four AE sensors
located in the driftway, was performed during the pre-stimulation characterisation phase. Each hit was also recorded
by a sensor fixed to the hammer, providing the origin time. These recordings were used to test the transmission of
elastic waves across the test volume and to obtain an estimate of the influence of deformation zones on the amplitude
and frequency content recorded by the AE sensors at varying recording distances. Together with the structural
analysis at the site, these measurements were used to determine final sensor placements of the seismic monitoring
system, omitting high-attenuation and deformation zones.

Similar active measurements were repeatedly performed at 24 fixed points in the vein drift and the driftway before,
during and after all other phases of the experiment (Figure 3) using sledge hammer and centre punch tools. To obtain
origin times for some of these hits, an additional accelerometer was installed next to the hitpoint. Centre punch tools
generate a more repeatable signal than the sledge hammer, with a defined impact force controlled by the internal
springs. We used three different centre punches with spring forces adjusted to 50N, 130N, and 250N. The spectra of
the generated impulse signals partially overlap with the spectra of AE events, containing higher frequencies
compared to the hammer impulse (Supplement material Figure S1). These hits, recorded by all AE sensors and
accelerometers, form an extensive dataset for AE sensor calibration, site attenuation and are a pre-requisite for
estimating magnitudes of the AE events (Kwiatek et al., 2011).

Sledge hammer hits also served as a simple reference signal to mark critical monitoring periods during all phases of
the experiment: Three hammer hits before the start and three to six hits at the end of each hydraulic pumping
operation allowed to calibrate timing of the seismic and hydraulic observation systems, made different groups on site
(located in different galleries during the stimulation) aware of operations and helped to distinguish working noise
from the target AE signals.



In addition to the active surveys along the tunnel walls, >300 ultrasonic transmission (UT) measurements were
performed in the hydraulic monitoring, injection, validation, and cable boreholes for velocity model estimation. The
ultrasonic transmitter (central frequency ~15 kHz) discharged a delta pulse of 7 µs duration. A total of 1024 of these
pulses were automatically stacked on each sensor channel to improve the signal-to-noise ratio. The resulting signal
generally contains more high frequency energy than common AE signals (>30 kHz, Supplement material Figure S1).
UT measurements in the injection borehole, with sources placed every metre along most of its length, were
performed for velocity measurements before and after the stimulation. The ultrasonic transmitter was placed in three
different orientations before the stimulation and at one orientation after the stimulation. The vertical validation
borehole was also sounded before and after stimulation, while the remaining validation and cable boreholes were
sounded once at the end of the validation phase or the final hydraulic testing phase of the experiment, respectively
(Figure 3).

**2.7    Passive seismic monitoring**
To monitor injection-induced fracture processes and associated small-scale brittle rock failure, we focussed passive
seismic monitoring on small magnitude ($M_W \leq -1.5$), high frequency ($f \geq 300$ Hz) AE events with expected fracture
sizes ranging from a few cm to the m-scale (Bohnhoff et al., 2010). Similar monitoring was previously successfully
applied (see review by Manthei and Plenkers, 2018; Kwiatek et al., 2018; Villiger et al., 2020).

Passive seismic (continuous and triggered) data were recorded during all injection operations. Triggering levels were
adjusted during hydraulic pumping operations and tuned for each stimulation interval to minimize false triggers that
lead to a dead time in the triggered recording system. Noisy channels were switched off to facilitate monitoring of
many partly overlapping AE events in real-time on site and to identify larger events. AE events detected in trigger
mode were automatically picked and located in near-real time on-site to obtain a pre-liminary catalog and control the
experiment. Outside of stimulation campaigns, the continuous-mode system was operated between 29 June and 14
August 2018 (with some data gaps, see Supplementary Material Table S1) and 5 November to 4 December 2019 (no
gaps) to measure post-stimulation processes and to characterize potential background seismicity. We recorded >72
TB of seismic data by the end of the field experiment.
**3    Methods**
**3.1    Data processing**
The different phases of the STIMTEC experiment were accompanied by varying in-situ noise conditions that
affected predominantly the high-sensitivity AE sensors. Passive seismic data often showed contamination with
transient electronic noise and noise generated by the hydraulic pumps during stimulation. To address this problem,
we applied filtering using the continuous wavelet transformation. We first identified the wavelet coefficients related
to transient noise signals by comparing continuous seismic data with and without noises. By removing the identified
wavelet coefficients from the recorded wavelet spectrum, the unperturbed AE signal could be retrieved efficiently.
This was possible because AE signal and noise overlapped only partially (Supplementary material Figure 2).

For post-processing of the triggered AE event data, we apply the automatic phase identification algorithm by Wollin
et al. (2018), which is based on the two-step approach by Küperkoch et al. (2010) to first determine a preliminary





arrival time, which is then refined by suppressing noise and using a wider causal band-pass filter. The waveforms are
first filtered using a third order Butterworth bandpass filter before a rolling higher-order-statistics kurtosis filter is
applied to determine a preliminary onset time. Then, by systematically calculating suits of Akaike's information
criterion (AIC)-functions on rolling and nested time windows of wavelet portions containing the phase onset, the
variability of the global minima is used to estimate the final pick as well as an asymmetric pick uncertainty.
Parameter settings are given in Table 3. The same procedure is applied for P- and S-arrivals. However, given the
single-component data and the AE sensor's typical post-pulse oscillations, automatically picked S-arrivals are
considered uncertain in this study. We observed that the amount of automatically picked S-arrivals is significantly
larger than for a reference dataset of manually picked S-arrivals. The reference dataset, comprising 300 events with
2,286 manual P- and 1,021 S-picks, was used to tune the automatic picking algorithm.

**3.2   Velocity model**
We used the active seismic UT measurements to derive a velocity model. UT data were manually inspected and
arrival times of the P- and S-waves, as well as the origin time of the UT source pulse, were identified. We
distinguished between impulsive, high-signal to noise ratio P-wave arrivals and more emergent, low-signal to noise
ratio onsets, with the latter being down-weighted by 50% for relocation and other procedures. Given the known
origin time and location of each UT measurement point, travel times to the seismic sensors were calculated assuming
straight ray paths. Uncertainties of the obtained velocities were assessed from repeated measurements from each
point in the injection borehole.

The Freiberg gneiss displays a prominent sub-horizontal foliation and was expected to show transverse isotropic
elastic properties as seen from core measurements (Adero, 2020) typically showing high P-wave velocities parallel to
the foliation and low P-wave velocities perpendicular to it. To describe the observed anisotropy of the obtained
velocity values, we applied the exact phase velocity equations for transverse isotropy (Thomsen, 1986, equations 10
a-d):
$$v_P^2 = v_{P0}^2 \left[1 + \varepsilon \sin^2\theta + D^*(\theta)\right],$$
$$v_{SV}^2 = v_{S0}^2 \left[1 + (v_{P0}/v_{S0})^2 \varepsilon \sin^2\theta - (v_{P0}/v_{S0})^2 D^*(\theta)\right],$$
$$v_{SH}^2 = v_{S0}^2 \left[1 + 2\gamma \sin^2\theta\right],$$
where $\varepsilon$ and $\gamma$ describe the strength of anisotropy for P-waves and for S-waves, respectively, $v_{P0}$ or $v_{S0}$ are velocities
along the symmetry axis, and $\theta$ is the phase angle. The parameter $D^*$ is defined as
$$D^*(\theta) = 0.5\left[1 - (v_{S0}/v_{P0})^2\right]\left\{\left[1 + 4\delta^* \sin^2\theta\cos^2\theta/(1 - (v_{S0}/v_{P0})^2)^2 + 4(1 - (v_{S0}/v_{P0})^2 + \varepsilon)\,\varepsilon \sin^4\theta\,/(1 - (v_{S0}/v_{P0})^2)^2\right]^{0.5} - 1\right\},$$
with:
$$\delta^* = (1 - (v_{S0}/v_{P0})^2)\,(2\delta - \varepsilon)$$
The angular dependence of the velocity is given by the shape factor $\delta$.
Using the full description is significantly more complex than the weak anisotropy approximation:
$$v_P^2 = v_{P0}^2 \left[1 + \delta \sin^2\theta\,\cos^2\theta + \varepsilon \sin^4\theta\right],$$
$$v_{SV}^2 = v_{S0}^2 \left[1 + (v_{P0}/v_{S0})^2\,(\varepsilon - \delta)\,\sin^2\theta\,\cos^2\theta\right],$$
$$v_{SH}^2 = v_{S0}^2 \left[1 + 2\gamma \sin^2\theta\right],$$





which was derived by Thomsen (1986) for weak-to moderate strength of anisotropy (ε, γ<0.2). This approximation is
commonly applied and describes the actual transverse isotropy accurately along and perpendicular to the symmetry
axis but not at intermediate angles.

We determined Thomson's anisotropy parameters for P-waves ($v_{P0}$, ε, δ) for each seismic station assuming full
transverse isotropy with a vertical symmetry axis. There was no angular asymmetry observed in the measured
velocities that would indicate a tilt of the symmetry axis. We assume that the recorded wave velocities represent
phase velocities rather than group velocities. We first calculated all wave velocities by systematically varying ε, δ in
steps of 2% and $v_{P0}$ in 100m/s steps. Then, the residual between computed and measured P-wave velocities were
computed in a comprehensive grid search over the sampled parameter ranges. Due to the scarcity of S-wave
observations in the UT data, the ratio of P-to-S wave velocities ($v_{P0}/v_{S0}$) along the vertical symmetry axis and the S-
wave velocity anisotropy parameter γ were fixed to 1.77 and 18%, respectively. These estimates were based on
Wadati (1933) plots for near-vertical ray paths and sonic logs from a 70 m-long, vertical borehole of the GFZ lab
(Giese and Jaksch, 2016). This sonic log shows the average value at shallow and deep depths, but a large deviation
for intermediate depths. The $v_{P0}/v_{S0}$ value is slightly larger than the average value obtained from the sonic log in the
(15°-inclined from horizontal) injection borehole. Both logs exhibit large scatter (±0.15). To determine the set of best
fitting Thomsen parameters per station (Table 4), we compared the parameter ranges for the best 10 and 100 models.
This velocity model was referred to as the best transverse isotropic velocity model per station. It was compared to an
isotropic velocity model ($v_P$=5600 m/s, $v_P/v_S$=1.76) and a single transverse isotropic velocity model for all stations
($v_{P0}$=5300 m/s, ε=11.3%, δ=0, $v_{P0}/v_{S0}$=1.76).

To clarify limits on the detection ranges as a function of distance, attenuation and anisotropy at the decameter scale,
we investigate attenuation characteristics of the rock. Attenuation estimates of the elastic waves travelling in the fast
anisotropy direction parallel to the foliation were obtained using hammer and centre punch hits. For each of the 10
hammer hits at each hitpoint along the galleries, an 8 µs time window starting at the P-arrival was chosen, from
which the maximum amplitude value was extracted. Then, the dominant frequency of the signal was determined for
each AE sensor from the maximum amplitude in the frequency range containing 99% of the energy of the signal. The
average of the dominant frequency from all sensors $f_{dom}$ together with the slope of the regression line m of the log of
the amplitudes with distance from the hitpoint and the average S-wave velocity $v_{S90}$ in the horizontal direction was
used to estimate the quality factor Q, according to:
$Q=|\pi f_{dom}/(m\ v_{S90})|$.
Also, attenuation estimates were obtained by comparing waveforms of centre punch hits recorded by accelerometers
located in opposite galleries with one sensor next to the hitpoint. Spectral ratios were analysed to obtain an estimate
of the quality factor.

**3.3    Hypocenter locations and velocity model uncertainty**
During post-processing hypocenter locations were determined using the equal differential time (EDT) method by
Zhou (1994) combined with a downhill simplex optimization algorithm (Nelder and Mead, 1965) applying the
developed transverse isotropic velocity model derived for each station. The EDT method has the advantage that the





inversion of the hypocenter location is based on the relative arrival-times of pairs of P- and S wave arrivals at the
same station or pairs of P-arrivals at different stations. The origin time is not specifically inverted for, but obtained as
a by-product. Gischig et al. (2018) demonstrated how the inversion for origin time, hypocenter location and station
corrections are affected by anisotropy. Applying the weak anisotropy approximation, these authors calculated the
velocity-dependent derivatives required for the inversion. We did not specifically account for anisotropy in the
location procedure, because the non-linear EDT method can handle 3-D heterogeneous velocity models. Instead, we
used the anisotropic velocities in the forward computation of the calculated travel-time grids, from which the EDT
surfaces were determined. We tested the method by relocating the known UT measurement points using the
manually identified P-arrival times with the derived velocity model per station.

To locate the AE events, we derived an initial hypocentre location based on P-wave arrivals only and a final location
including only those S-arrivals, consistent with the initially-derived hypocenter. To be included in the location
procedure, the root-mean-squared (rms) residual for an S-arrival needed to be less than 1.5 times the rms of the P-
arrivals for the initially derived hypocenter ensuring that inaccurate autopicked S-arrivals were discarded. The rms is
defined as
$rms = (\sum_i (t_i^{calc} - wt_i^{obs})^2 / \sum_i )^{0.5}$,
where $t_i$ are calculated and observed travel times for i stations and w is the weight. Phase weighting for autopicked P-
arrivals was implemented, based on the signal-to-noise ratio (SNR), with SNR≥6 obtaining full weight, 6>SNR≥3
half weight and SNR<3 one tenth of the full weight. S-arrivals were weighted with two tenth of the full weight if
included in the hypocenter estimation. We consider only events with a minimum of five phase arrivals and display
those hypocenter locations with rms travel time-residuals below a selected limit of 2 ms. We also applied station
residuals obtained as average P-wave travel-time residuals per station.

To assess the influence of the applied velocity model on the hypocenter locations, we compared the median rms
travel-time residuals of all AE event hypocentres obtained using different velocity models as well as the location
uncertainty of the relocated UT measurement points. By comparing the relocation error from the isotropic velocity
model with the transverse isotropy model and the best transverse isotropic velocity model per station, we provide
estimates for the location uncertainty associated with inaccurate velocity models.
**4    Results**
**4.1    Constraints on velocity models and location uncertainty**
Using a transverse isotropic velocity model per station, we obtained more accurate locations (lower rms travel-time
residuals, Table 5) and reduced the uncertainties determined from re-locating the known UT measurement points
(using the manually identified arrival times and the derived $v_P$- and $v_S$-velocities, Figure 4) compared to using an
isotropic velocity model or a single transverse isotropic velocity model for all stations. The latter was determined
from the averaged Thomsen parameters of all stations (Table 5). The best velocity model per station results in an
average relocation error of 0.26±0.06 m for the active seismic UT measurement points in the range 22–31 m
borehole depth in the injection borehole (Figure 4b), along which the majority of AE events were observed,
compared to 2.6±0.20 m for isotropic and 0.49±0.12 m for the single transverse isotropic velocity model. Relocation



of the UT measurement points was based on using only P-arrival times. Adding the S-wave arrivals did not further
reduce the location errors. This is likely because there are only few S-picks (on average 3 per measurement point for
the injection borehole and 5 for the vertical validation borehole, compared to on average 12 and 13 P-picks)
identifiable in the UT data. Note that the S-wave velocity model is not well constrained, but the few S-arrivals
observed in the active UT dataset are consistent with the assumed S-anisotropy parameters ($v_{S0}$, $\gamma$).
The best transverse isotropic velocity model per station also provided the lowest relocation error on average along
the injection borehole outside the damage zone (borehole depths <42m), where the resolution accuracy is decreased
by 70% for the isotropic model and 29% for the single transverse isotropic model (Table 5), respectively. We
observe that the best velocity model per station is tuned to the injection borehole because its number of measurement
points is largest. For relocating the known UT measurement points in the vertical validation borehole, relocations
obtained using the single transverse isotropic model (average relocation error of 0.69±0.53 m, Figure 4b) are more
accurate than for the best velocity model per station (average error 0.95±0.46 m). The isotropic velocity model
performs best in relocating the known UT measurements in the deformation zone based on the relocation error,
compared to the anisotropic velocity models. Within the deformation zone, all models show a systematic mislocation
upwards above the injection borehole (Figure 4a).

### 514 4.2 Structural heterogeneity, velocity and attenuation

We investigated the influence of the various geological structures in the rock volume on the seismic wave
propagation and on the velocity model. The background anisotropy caused by the strong foliation of the host rock is
overprinted by structural heterogeneity on site. We observed significant velocity reductions of 1-4% per station over
several UT measurement points (Figure 7a) associated with a prominent fault, identified at 32.5 m in the injection
well (Figure 2). We also see significant misfit between the velocities predicted by the anisotropic velocity model and
the observed velocities for deformation zones at borehole depths >42 m in the injection borehole and >32 m in the
long validation borehole (Figure 3). At these depths the logged structures and elevated fracture densities likely affect
seismic wave propagation by strong attenuation and deviating ray paths. This suggests that the velocity models
fitting the anisotropic reservoir rocks are inadequate for fault and surrounding damage zones.
Close to the prominent fault at 32.5 m depth, we observe an amplitude reduction of the stacked UT signal by about
50% compared to the values of neighbouring measurement points. This value was determined as the difference
between the actual value and the value expected for these depths from linear regression of neighbouring amplitude
measurements. Still ambiguity prevails as other factors such as UT source coupling and resonances at the receivers
can also affect the recorded amplitudes. In general, we do not observe a systematic velocity or amplitude reduction
from UT measurements in the injection borehole after stimulation as compared to before. Attenuation estimates of
the elastic waves travelling in the fast anisotropy direction parallel to the foliation obtained from hammer and centre
punch hits, resulting in $Q_P$-factors of about 50 near the galleries and 150 in the centre of the rock mass.
We observed good SNR ratios for UT measurements in the records of the three accelerometers for distances ≤15–18
m. For both accelerometers located off vein drift, we observed clipping of active centre punch hits generated at 10–



15 m distance with incidence angles around 90° to the accelerometer axis. This likely reflects resonances and/or
coupling issues. UT measurements are not recorded beyond distance of 31 to 33 m by the AE sensors. The AE-
hydrophone recorded UT signals with good SNR for distances smaller than 17 m (c.f. Boese et al, 2021). This
reduced recording range compared to the AE sensors is likely related to the impedance contrast of the water-filled
borehole and the rock. For this reason, AE-hydrophones need to be placed as close as possible to stimulated
intervals, or, alternatively, installed permanently by cementation, which reduced the impedance and increases the
sensitivity.

## 4.3    Seismic monitoring and network sensitivity improvements

Hydraulic stimulation started in the deepest part of the 63 m-long injection borehole with an intended progression of
stimulation from deep to shallow intervals (Figure 1). No AE activity was observed during stimulation of the two
deep intervals at 56.5 m and 51.6 m borehole depth, closest to the highly fractured damage zone encountered at the
bottom of the borehole. These intervals locate furthest from the seismic monitoring network (HF1 and HF2; Table 2).
To test detection limits and the seismic monitoring equipment under the given noise conditions, we changed the
intended order of the stimulated intervals, so that two shallow intervals (at borehole depth smaller than 30 m: HF3
and HF4) were stimulated next, followed by two intermediate depth intervals (borehole depth between 30 m and 45
m: HF5 and HF6) before returning to the deep intervals (borehole depth greater than 45 m; HF7 and HF8). We
observed significant AE activity (several thousand events, Table 2, Figure 5 and Figure 6a) for the shallow
stimulation intervals (22.4 m, 24.6 m, and 28.1 m) and high breakdown pressures (11–13 MPa). Seismic activity was
not identified before the start of the stimulation and stopped after shut-in. Few AE events were recorded during
injection into intermediate-depth intervals (33.9 m, 37.6 m and 40.6 m depth, Table 2). These events occurred
diffusely throughout the pumping sequence (Figure 6b). For the interval 33.9 m the second lowest breakdown
pressure (6.4 MPa) of all tests was observed, whereas the adjacent interval 37.6 m exhibited the highest observed
value (15.8 MPa), pointing towards significant spatial complexity. The breakdown pressure of interval 40.6 m (9.4
MPa) is comparable to those in the deep intervals 49.7 m, 51.6 m, 55.7 m, 56.5 m, which show intermediate to low
values (5.8–9.4 MPa, Table 2) and no AE events, neither during the stimulation nor during subsequent hydraulic
testing phases of the experiment.

For stimulations of the seismically active intervals in the injection borehole (HF3, HF4, HF10; Table 2) and in the
vertical validation borehole (HF12-15; Table 2, Figure 5), we observed a general correlation between seismicity,
fluid-injection cycles and volumes, when the injection pressure exceeded the fracture opening pressure. A small
number of AE events occurred during the frac and refrac sequences (5–70 AE per sequence), whereas significantly
more events were observed during subsequent step-rate tests (75–180 AE above jacking pressure) and during
periodic pumping tests (30–240 AE per cycle, Figure 6). We observed a progressive growth of the seismic clusters
which extend about 5 m radially from the injection interval (Figure 5). The sub-horizontal foliation does not seem to
noticeably influence event propagation and seismic cloud growth. Note that the seismic clusters from the injection
and vertical validation borehole are spatially distinct.



The highly variable seismic response to stimulation prompted us to relocate two of the 16 seismic monitoring sensors
(Figure 1) to test if the absence of AE activity results from limitations in network sensitivity or site characteristics.
We placed one AE-hydrophone at the bottom of the hydraulic monitoring borehole to verify AE detection levels for
intermediate-depth and deep stimulated intervals in the injection borehole. The AE-hydrophone recorded few AE
events during further hydraulic testing and accidental re-stimulation of the intervals 37.6 m and 40.6 m (at 6–8 m
hydrophone-interval distance), respectively, but no activity was observed for intervals 49.7 m and 56.5 m (at 10 m
and 17 m distance), confirming previous observations of no AE activity in the deep stimulation intervals. The
borehole broadband sensor was moved to the bottom of the vertical validation borehole for the last phase of the
experiment, so that it located at a comparable absolute depth as the deepest stimulated intervals in the injection
borehole. This was considered beneficial because of indications that seismic wave attenuation perpendicular to the
foliation may be larger than parallel to the foliation (Adero, 2020). Overall, the broadband sensor recorded
characteristic signals during hydraulic stimulations of all intervals in the injection borehole on 16-18 July 2018
(Figure 7 and Supplementary Material Figure 3) that resemble the injection pressure rather than the flow rate. These
signals were not recorded by the only other nearby broadband sensor FBE (SX Net, distance 438 m SE of STIMTEC
site). The observed signals vary in amplitude and period and are best observed on bandpass filtered (0.001–1 Hz)
daily seismograms on the second horizontal component of the ASIR sensor, likely aligned parallel to vein drift
(perpendicular to the borehole). There are also spike signals observed that may indicate rapid tilting and recalibration
of the sensor (see also Supplementary material Figure 3), based on shake table calibration after the experiment. They
occur during operations at the site and their interpretation currently remains unclear.
**5    Discussion**
**5.1    Seismic monitoring and network adaptions**
Using a seismic monitoring system consisting of AE-hydrophones, AE sensors, accelerometers and broadband
sensors bears several advantages. The AE-hydrophone can be attached on hydraulic tubing and therefore installed in
combination with hydraulic equipment. This places it much closer to the stimulated intervals and as a consequence,
AE-hydrophones can enlarge the 3-D density of sensors and their coverage in the volume of interest, thus improving
location accuracy and focal mechanism determination. AE-hydrophones do not require coupling to the rock mass and
are more easily installed than AE sensors. This comes at the cost of reduced recording ranges and frequency
bandwidth compared to common AE sensors (and reduced S-wave sensitivity cf. Boese et al., 2021).

All dedicated seismic monitoring boreholes were located above the stimulated volume to ensure that water entering
into the boreholes can drain during the experiment. This posed the general problem of increased location uncertainty
in the vertical direction. However, with this setup we achieved the desired monitoring quality without needing an
extra monitoring borehole placed close to the stimulation borehole. During the EGS Collab and GTS experiments,
the intersection of growing fractures with nearby monitoring boreholes caused immediate pressure release, inhibiting
fracture growth (Schoenball et al., 2020). This illustrates the problem that monitoring boreholes may impinge on the
stimulation. Therefore, high sensitivity AE sensors placed at some distance (20–30 m, considering the site
characteristics of the STIMTEC experiment) to the stimulated intervals combined with AE-hydrophones placed close
to the stimulated interval in the stimulation borehole (above the double packer) likely offer the best solution for high-



resolution seismic monitoring during hydraulic stimulation in URLs. However, preservation of the high-frequency
content of seismic waves is site dependent and a prerequisite for the analysis of source properties of AE events with
expected fracture sizes at the dm-scale (e.g. Kwiatek et al., 2011). Empirical results of Plenkers et al. (2010) provide
upper bounds for detection limits of AE events in low-attenuating hard rocks at ~3 km depth. In the more general
case, we refer to the modelling of detection limits for high frequency energy of microseismic events by Kwiatek and
Ben-Zion (2016).
Adapting the stimulation on site by changing the stimulation order in the injection borehole allowed for testing the
sensitivity of the monitoring system and site conditions but also resulted in the stimulation of the most seismically
active intervals (HF3, HF4, HF10; Table 2, Figure 5) on three subsequent days. This adaption was possible because
of the near real-time processing and visualisation of AE events on site. It allowed us to separate the temporal
distribution of the AE events in the spatially overlapping seismicity clouds (Figure 5).

### 5.2    Seismic response to stimulation

We observed significantly different seismic and hydraulic responses of intervals separated by only a few meters in
heterogeneous, metamorphic rock (Figures 5 and 6). This generally agrees with observations from the AHRL, GTS
and EGS Collab experiments, which also highlighted the influence of the rock type, the pre-existing fracture zones,
and stress heterogeneity on seismic responses to hydraulic stimulation. Although it is not yet clear what causes the
large variability in deformation behaviour at the STIMTEC site, we verified that it is not the result of detection
capabilities of the seismic monitoring network along the injection borehole. We posit that deformation in response to
stimulation in the deepest part of the injection borehole is predominantly aseismic, based on the absence of AE
events and the strong long-period signal recorded by the broadband sensor. We suspect the observed variability in
seismic response to stimulation is likely caused by rock-mass heterogeneity and the response of pre-existing
fractures. In addition, injection boreholes not aligned with one of the principal stress axes show complex fracture
initiation (Rummel, 1987; Haimson and Cornet, 2003), likely controlled by small-scale material heterogeneities at
the borehole wall, as also observed in lab experiments (Masuda et al., 1993). Reorientation of fractures with growth
away from the injection interval has been observed previously in boreholes misoriented with respect to the principal
stress axes, for example by mine-back in soft volcanic rock (Warren and Smith, 1985) and by AE event cluster
orientations in crystalline rocks (Gischig et al., 2018; Schoenball et al., 2020) and salt rock (Manthei et al. 2001). Re-
orientation of AE event clouds has not yet been identified during the STIMTEC experiment. We note, however, that
unexpected (based on stress modelling), strong, local variations of the stress magnitudes in the experimental volume
were obtained from direct stress measurements in the injection and vertical validation boreholes (Adero, 2020). The
variability of shut-in pressures (with the largest deviations from the average values observed in the adjacent
stimulation intervals at 33.9 m and 37.6 m depth in the injection borehole) and orientations of induced fractures
suggest overall small-scale stress heterogeneity at the STIMTEC site (Adero, 2020).
The observed low-frequency broadband recordings are similar to those broadband records observed by Zang et al.
(2017, their Figure 11) at the AHRL. In particular, we obtained strong signals from stimulations that did not yield
high frequency AE events. However, not all observed pressure peaks can be correlated with peak amplitudes of those



low-frequency seismic signals, suggesting that there seems to be a complex relationship, dependent on pressure
amplitude and period that requires further investigation. Our observations suggest that borehole sensors sensitive in
the frequency range 0.01–100 Hz positioned at distances of 19.6–26.6 m are adequate to monitor low-frequency
deformation associated with hydraulic stimulations.
**5.3    The role of anisotropy and heterogeneity for mine- and lab-scale experiments**
Laboratory and active seismic measurements from the STIMTEC experiment show moderate to strong elastic wave
anisotropy controlled by the pronounced foliation of the gneiss. We compare the here obtained Thomsen parameters
to those values determined in a range of laboratory tests on cylindrical Freiberger gneiss samples at different
confining pressures (≤30 MPa) and orientations at room temperature (Adero, 2020). P-wave velocity measurements
on samples in the laboratory exhibit similar mean values and ranges for wave propagation in different orientations
with respect to the foliation as observed in field measurements (Figure 8). This is irrespective of the significant
differences in frequency bands of UT sources in the laboratory (500 to 800 kHz) and in the mine (5 to 60 kHz). At
the STIMTEC site, P-wave velocities for ray paths parallel to the foliation are on average 12% higher than
perpendicular to the foliation for UT data. In laboratory tests, P-wave velocities for ray paths parallel to the foliation
are slightly larger, about 20% higher compared to a direction normal to the foliation (Figure 8, Table 1). A large
amount of active UT field measurements was needed to cover the range of incidence angles necessary to determine
the degree of P-wave velocity anisotropy and the symmetry axis of the metamorphic rock (Figure 3). Near-vertical
ray-paths (parallel to and at acute angles to the symmetry axis) were difficult to obtain due to geometrical constraints
limiting sensor positioning. In general, we observed a trade-off between the obtained P-wave velocity along the
symmetry axis $v_{P0}$ and the P-wave anisotropy parameter $\varepsilon$ for the UT data (Supplementary material Figure 4). This
likely is an effect of missing constraints near the symmetry axis because of few near-vertical ray paths for the
majority of stations. The two stations located furthest above the injection borehole with the highest number of near-
vertical incidence angles, display intermediate $\varepsilon$ values of 8–12% and $v_{P0}$=5200-5400 m/s. The average velocities of
$v_{P0}$=5275 m/s and $v_{S0}$=2980 m/s from a sonic log for comparable depths in the vertical borehole of the nearby GFZ
lab is consistent with the obtained velocity models. The average horizontal velocities of $v_{P90}$=5650 m/s and
$v_{S90}$=3260 m/s from sonic logs in the injection borehole at the STIMTEC site are lower than the average velocities
obtained for near-horizontal wave propagation from the UT data (Figure 8). These sonic log velocities are more
consistent with P-wave velocities derived for foliation-parallel wave propagation at the GFZ lab. We interpret the
lower values to reflect the effect of dispersion, given the frequency content of the measurement (4–30 kHz for sonic
log, 0.15–3 kHz for tomography at the GFZ lab, versus 5–60 kHz for active UT at the STIMTEC site).
Anisotropy complicates the analysis of all measurements in the STIMTEC test volume, especially regarding velocity
model calibration and AE event location. In retrospect, we estimate that approximately one third of all active UT
measurements in combination with the lab measurements, sonic logging and other available information (Krauß et
al., 2010) would have been sufficient to characterise the single transverse isotropic velocity model, which captures
the general features of the background anisotropy on site. This implies that the effect of dispersion is insignificant.
However, to resolve the best-possible velocity model for each station and to obtain high-accuracy AE event locations
required a transverse isotropic velocity model per station, derived from a large amount of active in-situ velocity



measurements covering a range of incidence angles. The best velocity model per station leads to a significant
location improvement of AE events from the injection and vertical validation borehole as shown by comparing the
rms travel-time residuals for different velocity models as well as the relocation error of known active UT
measurement points along the boreholes (Table 6). Neglecting anisotropy would lead to significant and systematic
location bias by up to 2.6 m (Figure 4b). The average P-wave anisotropy for the STIMTEC site is larger than
observed for the granite and granodiorite host rocks at the GTS (~7%) and AHRL but comparable to the phyllites at
SURF (Gao et al., 2020). Gischig et al. (2018) showed that at the GTS similar but slightly more scattered AE event
clouds could be obtained using the joint hypocenter determination method with an isotropic velocity model and
station corrections for AE event locations compared to using the anisotropic velocity model. Their approach is based
on the weak anisotropy approximation, but it suggests that the effect of anisotropy can be mitigated this way.
However, 32 seismic stations were installed at the GTS and structural heterogeneity is not as pronounced there as at
the STIMTEC site, because the shear zones are similar in orientation compared to the foliation causing anisotropy in
the rock volume. Our work demonstrates that high-resolution AE event locations (average rms=0.00015 s) can be
obtained in heterogeneous rocks with pronounced anisotropy, if an accurate velocity model can be derived. This
requires numerous UT calibration measurements from various angles, which is achievable for URL experiments,
some computational effort to derive the velocity model and a smart event location procedure. This demonstrates that
hydraulic stimulation in complex rock such as anisotropic and heterogeneous metamorphic gneiss is possible and can
be monitored (with additional effort), so future in-situ experiments do not need to consider homogeneous rocks only.
Lab experiments also documented a strong influence of the foliation on the mechanical strength and therefore on
fracture propagation and length (c.f. Adero 2020, Vervoort et al., 2014). The shallow depth of the STIMTEC site
results in low absolute stress magnitudes (1–6 MPa) and lower differential stress conditions compared to URL sites
elsewhere. To limit the effect of the foliation on the stimulation, the injection borehole was drilled at a 16°-angle to
the foliation. Despite the low absolute stress magnitudes, neither impression packer marks nor AE cluster
orientations indicated that the foliation determined fracture propagation in the injection borehole. This was also
found at SURF, where hydro-fractures did not follow the strong foliation but the inclined maximum principal stress
over tens of meters in the injection borehole (Oldenburg et al., 2016).
We observed significant velocity reductions (1–4%) associated with prominent pre-existing structures, in particular
in the deformation zones crossing the injection and long inclined validation boreholes (Figure 8). The amplitude
reduction of the stacked UT signal at these depth intervals could be 50%. In general, we do not observe a systematic
velocity or amplitude reduction from UT measurements in the injection borehole after stimulation as compared to
before. We conclude that only prominent pre-existing structures identified in logs have a significant effect (velocity
drop larger than the average measurement uncertainty) on velocity and attenuation. Whether transient fluid pressure
variations during the stimulation have a measurable effect on velocity (Doetsch et al., 2018) and/ or attenuation at the
STIMTEC site remains the subject of further investigations, which will be attempted using relative travel time times
from centre punch measurements as opposed to absolute travel times from UT measurements. P-wave attenuation
factors determined here for the fast anisotropy direction are generally consistent with the values obtained for the GFZ



lab (Krauß et al., 2010). Laboratory measurements revealed that attenuation perpendicular to the foliation is stronger
than parallel to the foliation (Adero, 2020), but this has not yet been investigated from the obtained field data.

**5.4    Implications for monitoring field-scale hydraulic stimulation experiments**

In field-scale projects, sensor placement is significantly more limited and constrained than in mine-scale settings,
where due to the 3D placement of sensors in close vicinity of the injection a close to ideal situation for monitoring of
a hydraulic stimulation experiment is achieved (similar to the laboratory scale). By avoiding permanent installations
and temporarily removing seismic sensors, we could use the existing boreholes for different purposes throughout the
STIMTEC experiment (e.g. for hydraulic monitoring, for passive seismic monitoring using different sensors, for
stress measurements, repeating measurements to verify impression packer marks and for repeated active seismic
measurements). Accessible boreholes provided us with more flexibility, especially as more boreholes became
available during the course of the experiment. Adapting the monitoring (by implementing, testing, and assessing a
new AE hydrophone and a broadband borehole sensor) and modifying the order of stimulations proved successful to
achieve the monitoring goals of STIMTEC. During a recent geothermal stimulation in Finland adapting the
stimulation procedure in response to high-quality real-time monitoring observations was critical for controlling fluid-
induced seismicity (Kwiatek et al., 2019). Maintaining flexibility during experiments at the mine and field scale,
which have less controlled conditions as compared to lab experiments, is a key element to address surprises and
unexpected challenges, which seem inevitable given the higher degree of reservoir complexity observed at these
scales. Flexibility requires good on-site communication between the various groups involved in the experiment, time
and budget to allow for changes, as well as practical and integrated approaches to manage, exchange, visualise and
interpret large 3-D data sets of different formats during the experiment.

Another observation of fundamental importance was that approximately half of the stimulated intervals were not
accompanied by any AE activity, despite appropriate monitoring in-place. Villiger et al. (2020) estimated the amount
of aseismic deformation during hydroshear experiments at the GTS and compared this to the amount of seismic
deformation, showing that aseismic deformation was dominant for both brittle and brittle-ductile structures. This
estimation was based on the total moment, calculated from borehole dislocations of mapped fractures, compared to
cumulative seismic moment of AE events and observed cloud extents. Guglielmi et al. (2015), De Barros et al.
(2019) and Cornet (2016, and references therein) also showed that deformation is mainly aseismic during
stimulations in softer rocks (shales, limestone) at the intermediate scale and sedimentary rocks at the field scale. To
simultaneously capture fast and slow deformation processes, which are currently often categorised as either seismic
or aseismic due to the limitations of current monitoring systems, requires better high-sensitivity instrumentation with
a wider bandwidth. Alternatively, the combination of sensors with different sensitivity and frequency ranges (e. g.
AE sensors, broadband, tilt, fibre-optic based strain sensors) is necessary, but requires time synchronisation and
amplitude calibration, which can pose sophisticated technical problems (c.f. Zang et al., 2017). To address these,
marker signals and regular active seismic measurements proved valuable during the STIMTEC experiment. The
mine scale has the advantage that new tools and/or different configurations (numerous sensor arrays) can be more
easily tested, and maybe regular high-resolution laser-scan tunnel mapping (Grehl et al., 2015) can be applied as an





equivalent tool to InSAR, which was successful in monitoring larger-scale slow- deformation processes at the
reservoir scale.
**6     Summary and conclusions**
Meso-scale experiments currently provide the most-detailed in-situ information to further understanding of hydro-
mechanical processes associated with hydraulic stimulation and allow for validation of inferred results. In the here
presented STIMTEC experiment, conducted in the Reiche Zeche mine URL at 130 m depth, we used a high-
resolution seismic monitoring network comprising twelve in-situ AE sensors (for high-sensitivity monitoring of
induced seismicity and the recording of active source signals), three accelerometers (for sensor cross-calibration
purposes), one broadband sensor (to extend monitoring to the low frequency range) and an AE-hydrophone (to
improve the network sensitivity in the deeper rock volume of the experiment). We relocated two monitoring stations
and tested new sensors during the course of the experiment to optimise passive and active seismic monitoring. In
contrast to other similar experiments, we stimulated strongly foliated rock with pronounced anisotropy during
STIMTEC. We acquired a large quantity of active UT measurements for characterising the anisotropy and
heterogeneity of the host rock. We monitored in near-real-time small-scale rock failure and friction processes
associated with hydraulic stimulation and tracked the spatio-temporal distribution of AE events.
Several key observations from the experiment are:
We demonstrated that high-frequency (up to 100 kHz) seismic monitoring in complex rock volumes with
pronounced anisotropy is possible, if measures are taken to accurately quantify the 3-D anisotropic velocity structure.
We applied Thomsen's exact phase velocity equations to deduce a transverse isotropic velocity model per station that
accurately locates known active ultrasonic measurement points in the stimulated boreholes. Estimates of
simplifications of the velocity structure and neglecting anisotropy significantly affect resolution and range between
0.5 and 2.6 m in our experiment.
We obtained average Thomsen parameters (P-wave anisotropy of 12%) in agreement with those derived from
laboratory and sonic logging data.
We observed that rock mass heterogeneity as seen in high-fracture density zones overprint the anisotropy of the host
rock and has a significant influence on velocity and attenuation.
We observed seismic responses to hydraulic stimulation in ten intervals in the injection borehole, performed with
similar injection protocol, ranged from abundant AE activity to no AE activity and are unrelated to monitoring
limitations. We attribute the observed variability in deformation to the small-scale rock mass and stress field
heterogeneity observed in the injection borehole.

Our observations indicate that stimulation of strongly foliated and fractured rock mass, such as the Freiberg gneiss,
results in activation of a complex fracture network. We infer that most of the induced deformation of the reservoir
remains aseismic given the high number of stimulated intervals with little or without AE activity and the observed
low-frequency signals recorded by the borehole broadband sensor. Aseismic deformation may be related to injection
into open pre-existing fractures in the injection interval; yet, borehole logs do not systematically show pre-existing
fractures present in 'quiet' stimulated borehole intervals.





**Data availability**

All active UT transmission data used in this study, sensor coordinates, measurement recording times and positions as well as manually identified phase arrivals are available from the GFZ data server (https://dataservices.gfz-potsdam.de/panmetaworks/review/fe97aa96c11b9aebca07a838bcb37a99e659ab2280e4045905484df76ae959c1/)

**Author contributions:**

GD conceptualised the experiment and acquired funding for it, GD, GK and KP planned the experiment and its instrumentation, CB, TF, KP, FB and CJ conducted fieldwork, JS and FB helped CB with data curation. CB administered the project, formally analysed the data and lead the investigations. CB wrote the manuscript with feedback and reviewing by GD, GK and KP.

**Competing interests:**

The authors (except TF, KP and FB) state that they are currently employed at the same institution as the journal's chief executive editor Charlotte Krawczyk.

**Acknowledgements**

This project was funded by the BMBF, project number 03G0874C. We thank LfLUG for providing fault surface data created on a small scale, a mine layout of the Reiche Zeche mine complex and surrounding mines and a digital elevation model. Staff from GmuG Bad Nauheim and Mesy SolExperts, Bochum, are thanked for their field measurement contributions to this project. Support by Frank Reuter and his team of miners at Reiche Zeche is gratefully acknowledged. Discussions with Joerg Renner and Marco Bohnhoff helped to improve this manuscript.

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



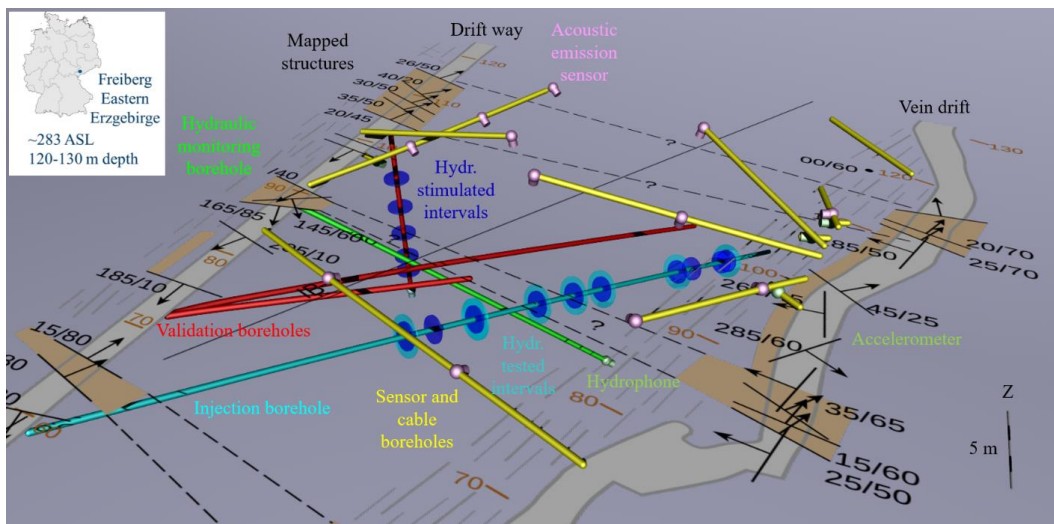



**Figure 1: Overview of the borehole network and mapped structures along the galleries at the STIMTEC site in the Reiche Zeche mine. Eastern gallery is the curved vein drift, the western gallery is the straight driftway. Deformation zones (brown zones) marked along the galleries and assumed to belong to connected systems between the galleries based on the orientations of mapped structures identified in the pre-characterisation phase of the experiment are shown. The monitoring system comprises twelve acoustic emission piezo-sensors (purple) located in horizontal or upward going seismic monitoring boreholes (yellow). Three accelerometers (light green) are collocated with AE sensors. A broadband sensor was moved from a short horizontal borehole off the vein drift to the vertical validation borehole (red) in driftway during the course of the experiment. An AE-hydrophone was placed at the bottom of the hydraulic monitoring borehole (green) for the last hydraulic testing phase of the experiment. Stimulation intervals (dark blue) in the injection borehole (cyan) and the vertical validation borehole (red) are shown together with repeatedly tested stimulation intervals (light blue). Inset shows the regional setting of the mine in Freiberg, Germany.**

1028

1029

1030



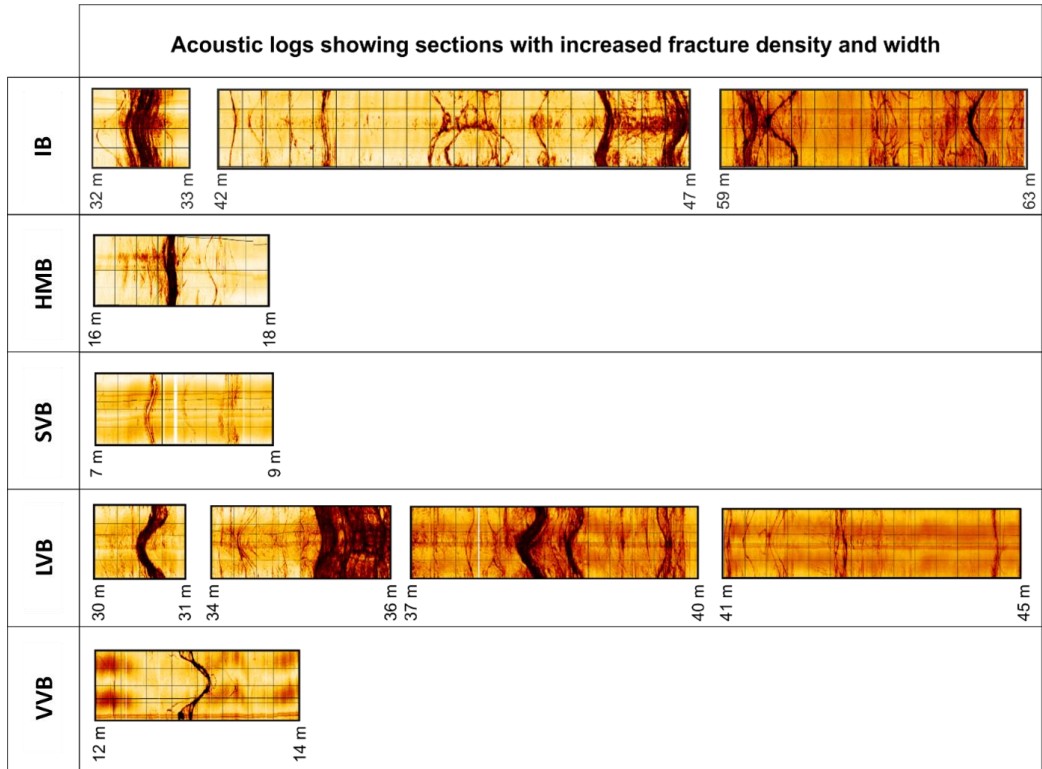

**Figure 2: Acoustic borehole televiewer logs indicating sections along the boreholes with increased fracture density and width, intercepted by the injection borehole (IB), hydraulic monitoring borehole (HMB), short inclined (SVB), long inclined (LVB) and vertical (VVB) validation boreholes. Modified from Adero (2020).**

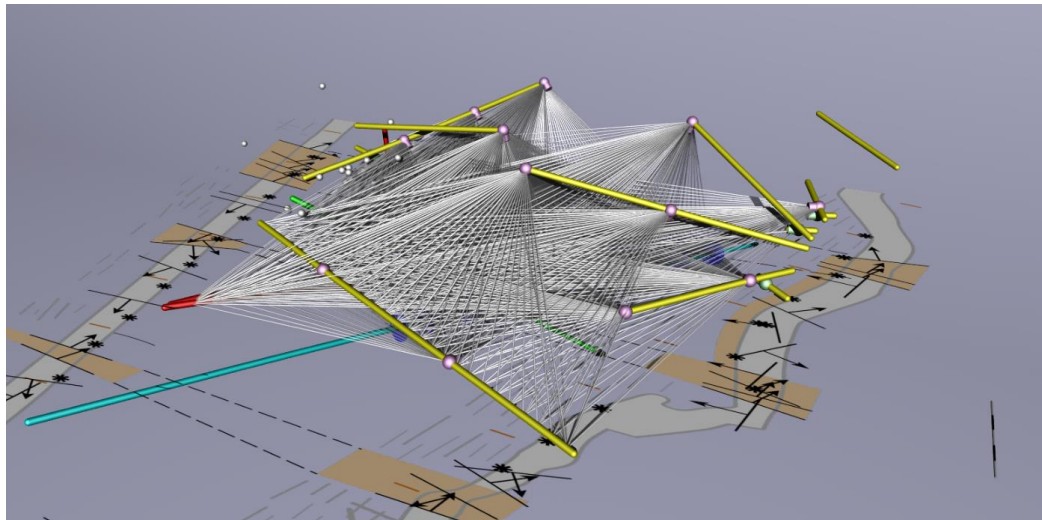

**Figure 3: Overview of active seismic measurements within the STIMTEC volume: Ray paths show coverage achieved using UT measurements from boreholes to sensors. Hitpoints (black stars) along the galleries mark positions of repeated active hammer and centre punch measurements.**



1040

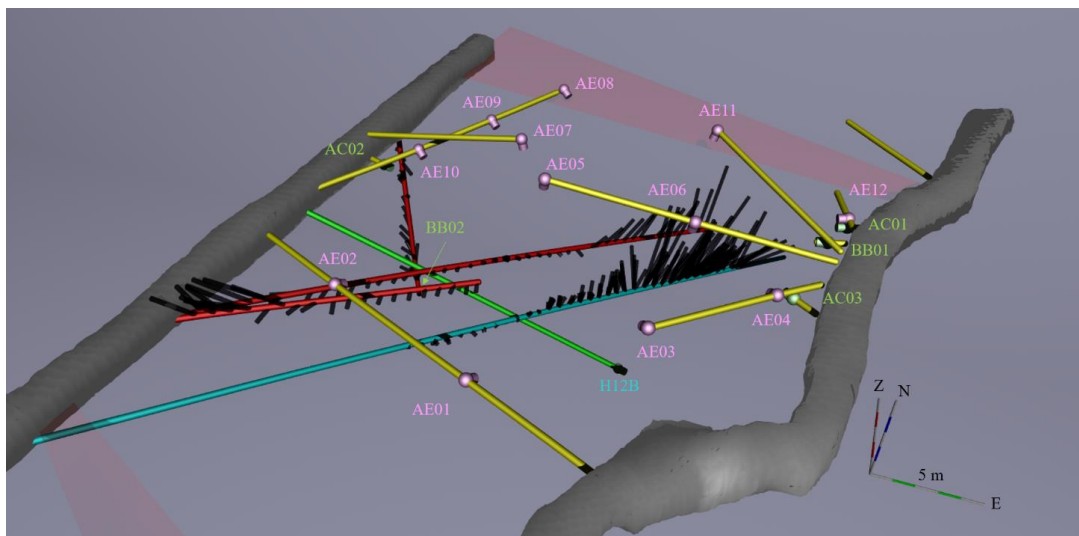

1041

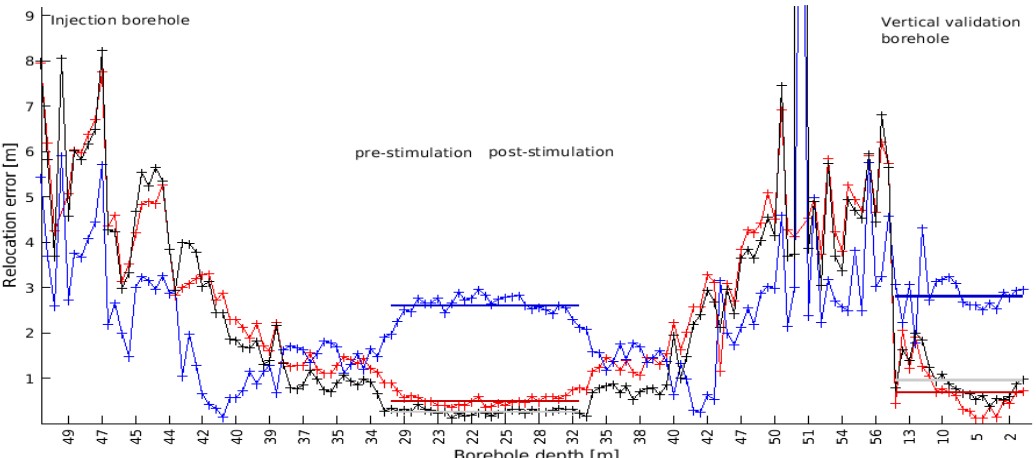

1042

**Figure 4: a) Overview of location uncertainty estimates (black lines) along the injection and validation borehole as estimated from locating known UT measurement positions (see Figure 3) with the derived best transverse isotropic velocity model per seismic sensor. Note that the location error becomes larger than 1 m, where the injection (cyan) and long inclined validation borehole (red) show higher numbers of fractures and more prominent ones (c.f. Figure 2). Labels refer to AE sensors (pink), accelerometers or broadband sensor (green, with the broadband sensor being moved to a new position during the experiment) and AE hydrophone (blue). Deformation zones (pink zones) that transverse the rock volume between the galleries are shown.**

**b) Comparison of location error of known active UT measurement points in the injection and vertical validation borehole for different velocity models. Relocation errors in black are obtained using the best transverse isotropic velocity model per station, in red from the single transverse isotropic velocity model and in blue from the isotropic velocity model. Coloured horizontal lines represent averages relocation errors for the given depth range.**

1054





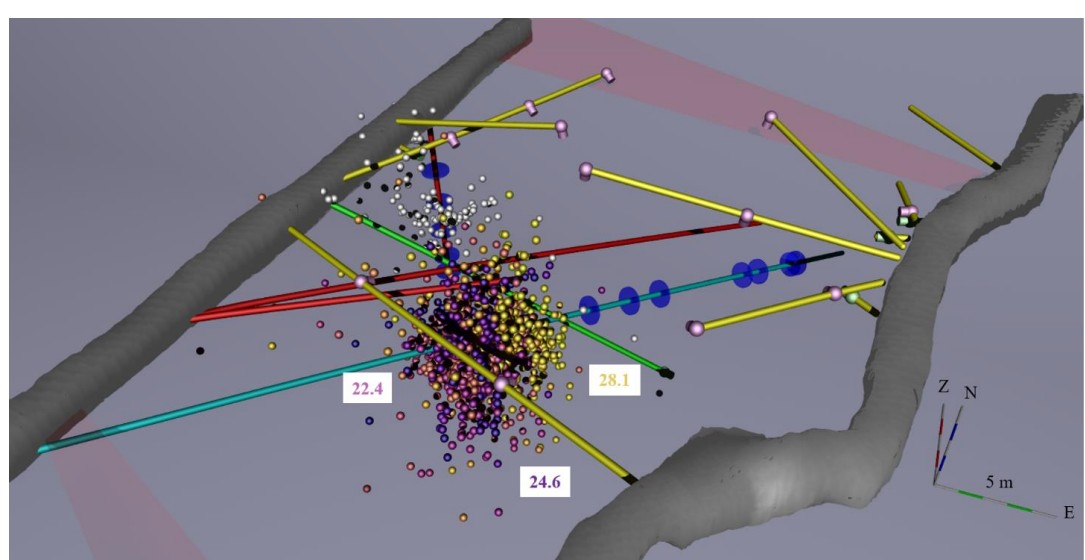

**Figure 5: Acoustic emission (AE) locations obtained for stimulation in the injection borehole (coloured dots according to stimulated interval as marked) and the vertical validation borehole (white dots). Note that the intermediate-depth and deep stimulated intervals in the injection borehole produced little to no AE activity**

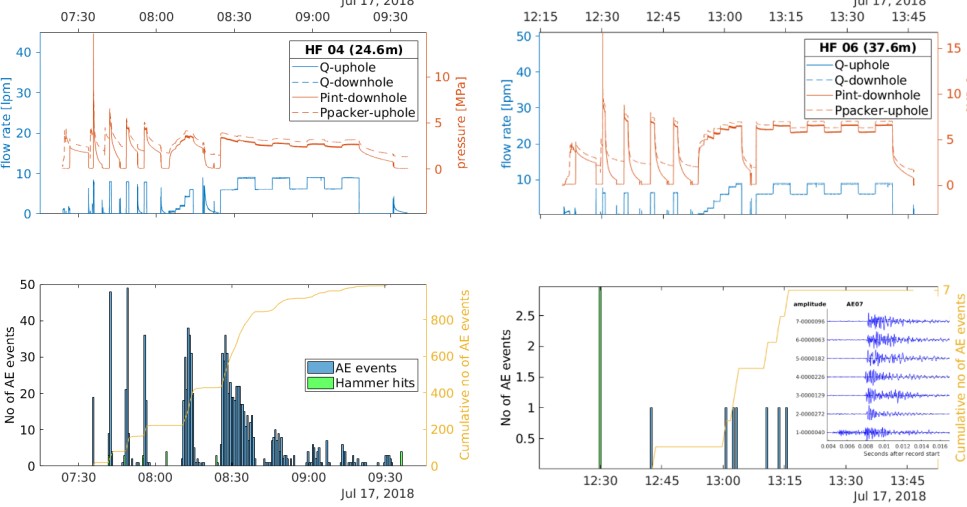

**Figure 6: Stimulation sequence consisting of a frac, three refracs, step-rate pump test and periodic pumping test for the intervals at 24.6 m and 37.6 m borehole depth in the injection borehole. Top panel shows flow rate (blue) and pressure records (orange) measured in the intact intervals downhole and at the wellhead uphole, bottom panel shows histogram (blue) and cumulative number (yellow) of located acoustic emission (AE) events. Active hammer hits (green bars) were used as marker signals for the beginning and end of the injection sequence. An example of all the AE events observed during stimulation of interval 37.6 m as recorded by sensor AE07 is shown as an inset.**





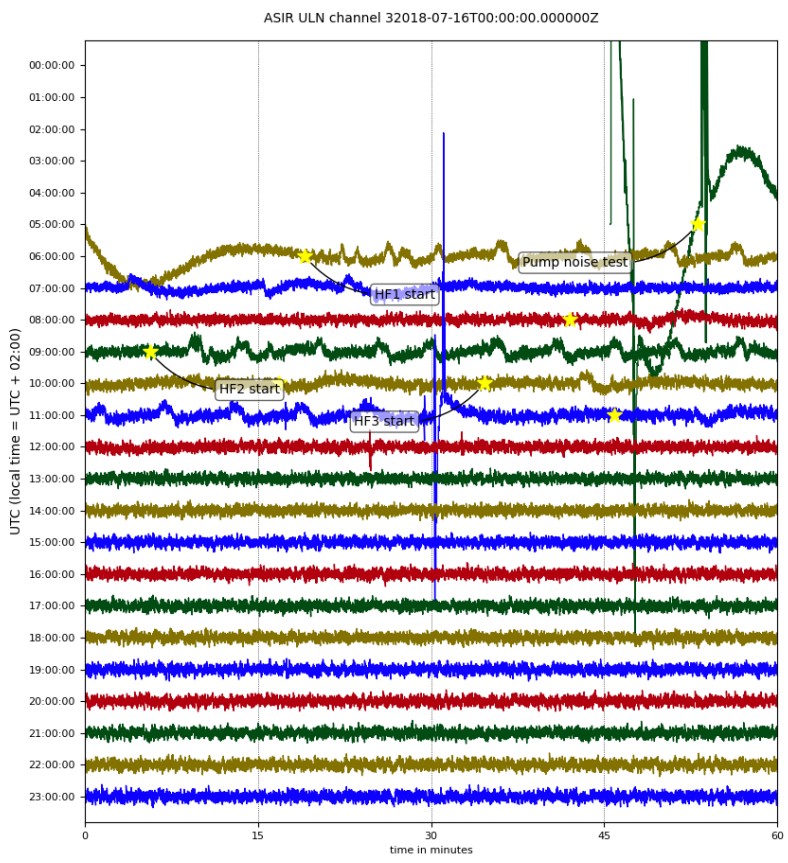

1070


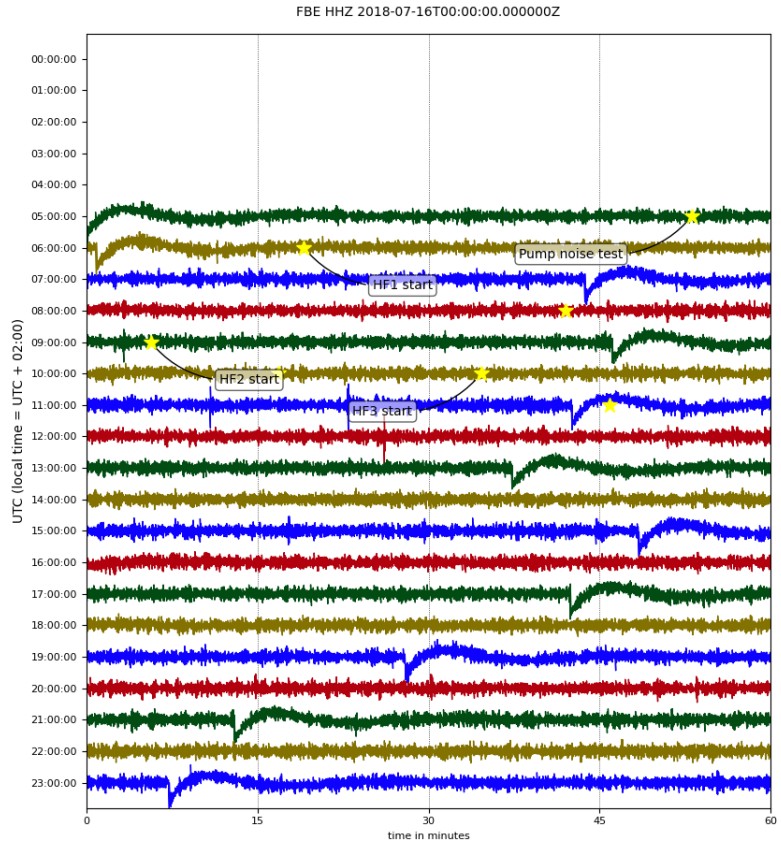

Figure 7: Daily records of the horizontal channel of the ASIR and vertical channel FBE broadband sensor located at Reiche Zeche mine for the first day of stimulation on 16 July 2018. The distance between both sensors is ca. 440 m. Hydrofrac start and end times are marked (by yellow stars and labelled at the start time) as listed in Table 2. Note that long period swings in the records result from bandpass filtering (0.001–1 Hz) in combination with data gaps as seen for the beginning of the records for the ASIR sensor and throughout the day for FBE. Some local quarry blasts are seen on both sensors, whereas stimulation related signals are only visible on the ASIR broadband sensor deployed at the STIMTEC site. Note that the two largest drops seen for the ASIR sensor are likely associated with sensor self-centering as determined on a shake table at GFZ lab after the experiment.


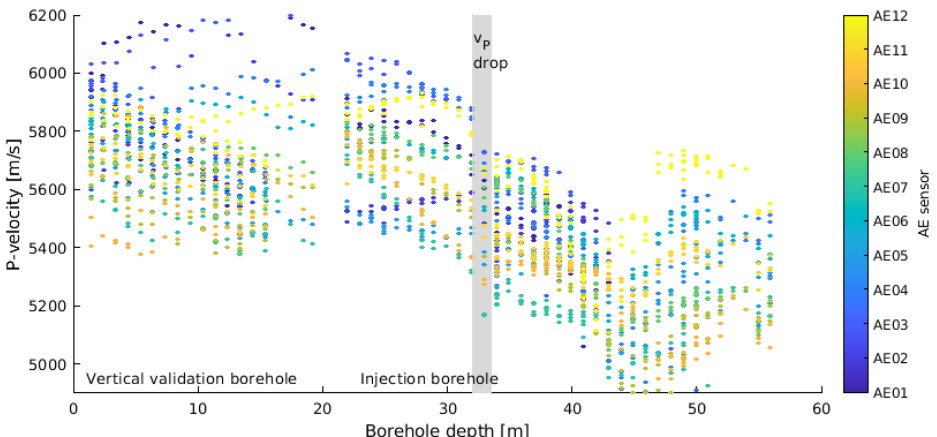

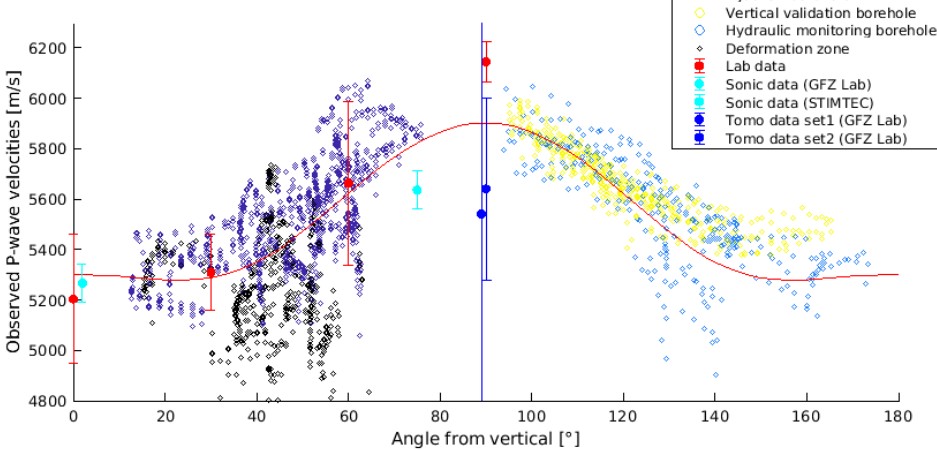

**Figure 8: Ultrasonic transmission velocity measurements used for calibration of the transverse isotropic P-wave velocity model shown for measurement depth in the borehole (top) and for angles relative to the vertical symmetry axis (bottom). The red circles display laboratory P-wave measurements (mean and standard deviation) on Freiberg gneiss samples and the red line the theoretical P-wave velocities with incidence angle as determined using the Thomsen parameters derived from the laboratory measurements. Measurement ranges obtained from sonic logs (cyan) from the vertical borehole in the GFZ lab and from the 15°-inclined STIMTEC injection borehole, as well as from P-wave tomography parallel to the foliation direction in the GFZ-lab (blue).**





**Table 1: Results of stress measurements through overcoring by Mjakischew (1987) at 140 m depth in the Reiche Zeche Mine**

| Principal stress | Magnitudes [MPa] | Orientation/Plunge [°/°] | |
|---|---|---|---|
| σ1 | 4.5 | 347/0 | NNW/Horizontal |
| σ2 | 3.6 | 0/90 | -/Vertical |
| σ3 | 3.0 | 77/0 | ENE/Horizontal |

**Table 2: Overview of stimulation details for the ten stimulated intervals of the injection borehole. Note that hydraulic characteristics (fracture open/closed injectivity and jacking pressure) were determined from the step rate test. The total injected volume and number of AE events are given for the whole stimulation sequence as shown in Figure 4. The stimulation intervals were chosen to contain as little pre-existing structures as possible based on cores and acoustic logs. The interval condition was reassessed based on the stimulation results as either intact where hydrofracs were created or pre-fractured, meaning that hydroshearing occurred.**

| Interval | HF10 | HF4 | HF3 | HF5 | HF6 | HF9 | HF8 | HF2 | HF7 | HF1 |
|---|---|---|---|---|---|---|---|---|---|---|
| Depth [m] | 22.4 | 24.6 | 28.1 | 33.9 | 37.6 | 40.6 | 49.7 | 51.6 | 55.7 | 56.5 |
| Date (2018) | 18/7 | 17/7 | 16/7 | 17/7 | 17/7 | 18/7 | 18/7 | 16/7 | 18/7 | 16/7 |
| Local time start | 10:50- | 07:20- | 12:35- | 11:15- | 12:20- | 09:40- | 08:50- | 11:05- | 07:40- | 08:20- |
| Local time end | 12:50 | 09:35 | 13:15 | 12:15 | 13:45 | 10:25 | 09:30 | 12:15 | 08:30 | 10:50 |
| Breakdown p [MPa] | 13.3 | 13.3 | 11.1 | 6.4 | 15.6 | 9.2 | 9.4 | 7.7 | 5.8 | 8.2 |
| Injected V [l] | 457 | 466 | 200 | 115 | 327 | 73 | 55 | 145 | 105 | 200 |
| mean sensor dist. | 19.5 | 18.7 | 17.8 | 17.7 | 18.5 | 19.5 | 24.6 | 26.0 | 29.1 | 29.7 |
| No. AE events | 4537 | 5775 | 867 | 6 | 8 | 1 | 0 | 0 | 0 | 0 |
| Pump. period [s] | 400 | 400 | 90 | 150 | 250 | – | – | – | 100 | 30–240 |
| Interval condition | intact | intact | intact | frac. | intact | frac. | frac. | frac. | frac. | frac. |

**Table 3: Minifrac measurement interval details for the vertical validation borehole. See Table 3 for explanation.**

| Interval | HF15 | HF14 | HF13 | HF12 | HF11 |
|---|---|---|---|---|---|
| Depth [m] | 4.0 m | 6.7 m | 9.3 m | 11.7 m | 13.2 m |
| Date (2019) | 21/8 | 21/8 | 21/8 | 21/8 | 20/8 |
| Local time start | 11:00- | 10:05- | 9:00- | 8:10- | 13:10- |
| Local time end | 11:45 | 10:46 | 9:45 | 8:40 | 14:00 |
| Breakdown p [MPa] | 11.07 | 14.95 | 7.95 | 14.73 | 7.46 |
| Injected V [l] | 22 | 19 | 21 | 18 | 33 |
| mean sensor dist. | 22.5 | 23.5 | 24.8 | 26.1 | 27.0 |
| No. AE triggers | 303 | 188 | 52 | 56 | 9 |
| Interval condition | frac. | intact | frac. | intact | frac. |





**Table 4: Parameter setting for automatic picking routine.**

| parameter | initial pick | final pick |
|---|---|---|
| 3rd order band-pass filter | [8, 50] kHz | [0.05, 120] kHz |
| AIC window width | 0.0015 s | same? |
| boundaries for uncertainty limits | [-0.0012, 0.0088] s | |
| window boundaries for mean energy | [-0.001, 0.009] s | |
| min SNR (amplitude/standard dev.) | (3,2) | |

**Table 5: Thomsen parameters (ε, δ, and ɣ) characterising elastic anisotropy of the rock mass derived from fitting all active seismic UT measurements per seismic station. The last two columns give the numbers of measurement points for vP and vS, from which the parameters were derived.**

| Station | $\varepsilon$ | $\delta$ | $v_{p0}$ | $v_{s0}$ | $\gamma$ (fixed) | Number $v_p$ | Number $v_s$ |
|---|---|---|---|---|---|---|---|
| AE01 | 0.02 | -0.10 | 5.8 | 2.9545 | 0.18 | 70 | 58 |
| AE02 | 0.02 | -0.18 | 5.7 | 3.2386 | 0.18 | 72 | 36 |
| AE03 | 0.02 | 0.14 | 5.5 | 2.5568 | 0.18 | 63 | 25 |
| AE04 | 0.02 | 0.20 | 5.9 | 2.5568 | 0.18 | 66 | 1 |
| AE05 | 0.08 | -0.01 | 5.4 | 3.0682 | 0.18 | 73 | 12 |
| AE06 | 0.16 | 0.38 | 5.8 | 2.6705 | 0.18 | 73 | 6 |
| AE07 | 0.12 | 0.14 | 5.2 | 2.9545 | 0.18 | 73 | 22 |
| AE08 | 0.28 | 0.84 | 4.6 | 2.5568 | 0.18 | 73 | 11 |
| AE09 | 0.14 | 0.04 | 5.2 | 2.8977 | 0.18 | 73 | 49 |
| AE10 | 0.1 | -0.16 | 5.5 | 2.6136 | 0.18 | 73 | 32 |
| AE11 | 0.10 | 0.26 | 5.2 | 2.8977 | 0.18 | 73 | 3 |
| AE12 | 0.02 | -0.22 | 5.9 | 2.7841 | 0.18 | 71 | 1 |
| AC02 | 0.04 | -0.18 | 5.5 | 3125 | 0.18 | 128 | 0 |

**Table 6: Comparison of root-mean-square residual and number of obtained event locations in the injection (IB) and vertical validation borehole (VVB) obtained using different velocity models. For location accuracy assessment, the average relocation error of the known UT measurement points outside of identified damage zones is provided which represents an average of all values shown in Figure 4b.**





| Velocity model | RMS IB $\cdot 10^{-4}$s (number AE events located with P and S) | RMS VVBH $\cdot 10^{-4}$s (number AE events located with P and S) | Average relocation error outside damage zones (m) (located with P only) |
|---|---|---|---|
| Isotropic model ($v_P$=5.6 km/s, $v_P/v_S$=1.76) | 2.8± 1.2 (2842) | 1.6± 1.3 (401) | 1.7±0.80 |
| Transverse isotr. model ($v_{P0}$=5.3 km/s, $v_{P}0/v_{S}0$=1.76, $\epsilon$ = 11.3%) | 2.9± 1.3 (3080) | 1.3± 1.3 (402) | 1.1±0.78 |
| Transverse isotr. model with SNR weighting | 1.9± 1.3 (4634) | 1.3±1.3 (405) | 0.9±0.65 |
| Trans. isotr. model per station ($v_{P0}$=5.25 km/s, $v_{P}0/v_{S}0$=1.76, $\epsilon$ = 12%) | 1.6± 1.2 (4613) | 1.0± 1.3 (395) | 0.8±0.73 |
| Trans. isotr. model per station with SNR weighting | 1.5± 1.3 (5531) | 0.9±1.3 (392) | 0.8±0.70 |