# Peer review of "Seismic monitoring of the STIMTEC hydraulic stimulation"

_Solid Earth, 2021_

## Referee Comment (RC1)

8-27-2021
Chet Hopp
Postdoc
Lawrence Berkeley National Lab

se-2021-84 Reviewer comments:

**Overall assessment:**

In "Seismic monitoring of the STIMTEC hydraulic stimulation experiment in anisotropic metamorphic gneiss", Boese et al. present an overview of the seismic monitoring network installed at the STIMTEC site in the Reiche Zeche underground lab, as well as a number of analyses focusing on wave propagation and seismic event location in the experiment volume. In particular, the authors focus on deriving a velocity model that accounts for anisotropic propagation speeds of seismic waves and the velocity model's effect on the location uncertainty of the detected acoustic emission events. The authors also do an excellent job of placing the STIMTEC experiment and results in the context of the growing body of underground lab work and the publication of these results will be a significant help to a number of researchers (myself included). The paper is very well written, logically organized, and worthy of prompt publication.

I've made a number of minor comments below, some of which are maybe more conversational than anything else. I do think the paper would benefit from a better illustration of the network geometry and distribution of events, as well as to more complete summary of the entirety of the STIMTEC stimulations. As it is currently, the paper focuses on the calculation of velocity model parameters with some asides to mention the relocation of a broadband sensor or to show the relationship between the rate of seismicity and hydraulic parameters for a couple of intervals. My suggestion would be to either 1) more completely document these tangential parts of the paper, including more clearly relating the broadband data to hydraulic data and documenting all of the intervals (not just the two shown in Figure 6), or 2) refocus the paper on determining the velocity model parameters and their effect on the AE locations. My personal preference would be for option 1) given that this will provide a robust foundation for further STIMTEC publications. I don't know what the STIMTEC manuscript-in-prep ecosystem looks like, so I, of course, leave that to the authors.

The paper was a pleasure to read and I look forward to seeing the revised version.

Chet

**General comments:**

The formatting of the equations is a bit hard to follow (D*, for example). I'll admit that even Thomsen needed two lines for that one.

The citation Renner et al., (2021) isn't yet available and is cited differently than in the author's ARMA paper. Is it a special issue to be published at the end of the year? Apart from conference proceedings, Renner is the only other potential publication detailing the STIMTEC experiment. Unless I'm simply missing another overview paper (entirely possible, and apologies if that's the case), then I think this current work needs to include some additional detail about the seismic network, making it the 'go-to' citation for further work (see comment on line 286).

Below, I also mention a need for better visualization of the network, seismicity, or (ideally) both. Something similar to the author's figures in the ARMA paper on the AE hydrophones (Fig 2, specifically) would help show the reader the vertical and map-view distribution of sensors and seismicity and confirm that there are indeed distinct clouds for the two stimulated boreholes.

**Specific comments:**

Ln 101: We're using Continuous Active Source Seismic Monitoring (CASSM) in our meso-scale experiments at LBL. Admittedly, the publications from URLs (Mont Terri or SURF) are only in conference proceedings so far, or not yet completely analyzed...But there are some cases of $CO_2$ plume imaging, for example: https://doi.org/10.1002/2017JB014164 Feel free to cite only if you think it's pertinent to the point you're making.

Ln 124: Here's the most recent analysis of the Collab seismic data, hot off the press: https://doi.org/10.1029/2020JB020840

Ln 187: I'm a little confused here. You're trying to maximize the angle between the foliation and injection borehole, correct?

Figure 1: Hard to show on an oblique view, but what's North? I may have missed this, but is the driftway oriented due North?

Ln 269: Previously you mentioned the EDZ is up to 10 m in width. How did you determine you had drilled beyond this with the 1.5 m holes?

Ln 286: Not strictly relevant to the analyses in this work, but since you're setting the stage for future analyses (and there appears to be no other review of STIMTEC published), a little more detail on the installation would help (selfishly, it will also help me with EGS Collab and other future experiments too!)
  • How are the GMuG sensors clamped to the borehole wall?
  • How are the Wilcoxons installed? Are they potted in some sort of housing? Are they also clamped? If they're potted, has the effect of the potting on the frequency/phase response of the sensor been characterized (these will be used as the ground-truth for the calibration of the AE sensors, correct?).

Ln 333: So all center punch shots (50, 130, 250N) were recorded on all accelerometers? Another selfish clarification, as I'm hoping to use this technique at EGS Collab.

Ln 335: Are the magnitudes still in the works and intended for a separate paper? I see a hint at relative magnitudes in the ARMA paper.

Ln 348: Just clarifying: The UT surveys comprised 1024 pulses at each meter along the injection borehole, correct? Is this a single transmitter, or a multi-level string of them?

Ln 350: I'm not familiar with UT sources. Can you please give a little more detail on what it is and how it's deployed? Also, what do you mean here by 'different orientations'? Is this because the UT sources have an irregular radiation pattern that you're trying to account for? Or do you mean different orientation as in, 'different position in the borehole'?

Figure 4a: I don't think you state anywhere what the direction of the black uncertainty vectors signify. Also, is it a coincidence that the more highly fractured zones at the top and bottom of the borehole correspond to the zones of higher uncertainty and also the zones where your array coverage is poorest (i.e. larger azimuthal gaps)? It's a little difficult to make out the network geometry with this one oblique view. Can you provide a map-view and a cross section along with one of the figures (e.g. Figure 1, 4a, or 5)?

Figure 4a: All of the larger uncertainties point upwards. Is this a product of such a one-sided array with only 1 sensor (AE hydrophone) below the stimulation intervals? Or, as you state, an underestimation of the attenuation in the fractures zones?

Ln 549: What did this reveal about the detection limits?

Ln 569: Can you also show the progressive growth with distance from the injection interval in Figure 6?

Ln 571-572: Figure 5 would be a good place for a map-view and cross-section to show the reader that the seismicity from injection and vertical validation are distinct.

Figure 6: A few comments:
- Text is too small in most cases
- AE and Hammer bars in the histogram are indistinguishable
- Very hard to see the waveforms, so what's being shown? You say 'all' events, so are these all the waveforms superimposed on each other, stacked, something else?
- What about the other seismically-active intervals?

Ln 586: It's hard to assess the claim of that these signals resemble pressure and not flow without adding the hydraulic parameters to these plots in some way.

Figure 7: The discussion of the broadband sensors seems a little tangential to the rest of the paper which I see as focused on the AE locations and velocity model. It might be better to leave this out or transfer to the supplements?

Ln 601: Plus no (good) S-wave

Ln 607: And destroyed some of the monitoring equipment…

Ln 609-612: Its a difficult trade-off between near-field monitoring and allowing the fracture to grow unimpeded. EGS Collab #2 will have a central injection well with four surrounding open boreholes (all drilled from same wellhead). Seismic instrumentation will be further afield this time in separate grouted wells (in an attempt to avoid what happened last time). Still, we'll likely intercept the fracture with one of the boreholes surrounding injection. As you mention, this significantly affects the fracture's behavior but also allows us to monitor the fracture at multiple locations (e.g. with a straddle packer or 3D displacement probe). But I agree that seismic monitoring should stay out of the way, especially when using high-sensitivity AEs.

Ln 648-652: If you choose to leave the broadband recordings in the main body of the paper, it would be nice to see a comparison of the waveforms and the hydraulic parameters. Not necessarily an interpretation of mechanisms, but a clearer view of the pressure-waveform relationship you mention in

the text. In addition, half of the waveforms in the Figure 7 helicorders are just noise and not adding a ton to the story.

**Technical corrections:**

Ln 79: EGS not defined beforehand

Ln 136: 'focusses'

Ln 195: I have a hard time following this sentence. Maybe one sentence per mine-back borehole?

Ln 220: '[the] vein drift'?

Ln 384: suit[e]s?

Ln 446-449: This is hard to follow. Please rephrase or break into smaller chunks.

Ln 523: [the] fault or fault[s]?

Ln 532: resulting → result

Ln 643: obtained or observed/measured?

Ln 725: time times

Ln 781: Should these be a bulleted list?

Ln 785-787: Unclear. "Estimates of the uncertainties related to simplifications...", maybe?

Ln 790: overprint[s]

Ln 793: protocol[s]

---

## Author Comment (AC1)

We would like to thank Chet Hopp (Reviewer 1) for his detailed review and for providing links to new, complementary publications. We have addressed every comment, suggestion and question raised by Chet and our answers are shown in **bold** below. All changes referred to in the manuscript are marked as tracked changes.

Overall assessment: 2nd paragraph

I do think the paper would benefit from a better illustration of the network geometry and distribution of events, as well as to more complete summary of the entirety of the STIMTEC stimulations. **We have added several figures to the Supplementary Material to better illustrate network geometry, raypath geometry and event distribution. All STIMTEC stimulations are listed in Table 2 and 3 to provide an overview and will be discussed in greater detail in another dedicated publication on the induced AE activity.**

My suggestion would be to either 1) more completely document these tangential parts of the paper, including more clearly relating the broadband data to hydraulic data and documenting all of the intervals (not just the two shown in Figure 6), or 2) refocus the paper on determining the velocity model parameters and their effect on the AE locations. **We have followed 1) because this publication is focused on the seismic monitoring of the STIMTEC experiment, which includes the lower frequency range recorded by the broadband sensor.**

General comments:

The formatting of the equations is a bit hard to follow (D*, for example). **Improved as much as possible, but these are complex equations. We extended the description on the determination of Q.**

The citation Renner et al., (2021) isn't yet available and is cited differently than in the author's ARMA paper. **It is the same Reference, which we have clarified more and also made all the changes suggested in comment line 286. All previous STIMTEC publications were conference presentations with proceedings and newsletter summaries.**

Below, I also mention a need for better visualization of the network, seismicity, or (ideally) both. **We have addressed this in several new figures in the Supplementary Material.**

Specific comments:

Ln 187: I'm a little confused here. You're trying to maximize the angle between the foliation and injection borehole, correct? **Yes, we restated this to make it clearer.**

Figure 1: Hard to show on an oblique view, but what's North? I may have missed this, but is the driftway oriented due North? **Yes, the driftway runs north-south. We added the 3-D coordinate system to this and the other figures.**

Ln 269: Previously you mentioned the EDZ is up to 10 m in width. How did you determine you had drilled beyond this with the 1.5 m holes? **We could not yet determine the extent of the EDZ at the STIMTEC site. We have clarified the sentence.**

Ln 286: Not strictly relevant to the analyses in this work, but since you're setting the stage for future analyses (and there appears to be no other review of STIMTEC published), a little more detail on the installation would help. **All points addressed where added to the text. Given that the mounting plate and the glue used to attach the Wilcoxons are both thin (total 3 mm) their effect on the accelerometers' frequency response is in the range beyond the frequency range of interest.**

Ln 333: So all center punch shots (50, 130, 250N) were recorded on all accelerometers? **No, as clarified in the text. Please note that I have a paper in preparation on the centre punch signals containing more details.**

Ln 335: Are the magnitudes still in the works and intended for a separate paper? **Yes, that's correct.**

Ln 348: Just clarifying: The UT surveys comprised 1024 pulses at each meter along the injection borehole, correct? Is this a single transmitter, or a multi-level string of them? **Yes, a single transmitter was used as clarified in the text.**

Ln 350: I'm not familiar with UT sources. Can you please give a little more detail on what it is and how it's deployed? **We added this in the text and provided more details on the orientations and the radiation pattern. The UT used by us is an AE piezo that is operated in the "reverse" way than for recording AE events (powered to generate a displacement). It is installed/coupled exactly the same way as the AE sensors.**

Figure 4a: I don't think you state anywhere what the direction of the black uncertainty vectors signify. **We have added a statement on the direction of the black uncertainty vectors in the first paragraph of Section 4.1 and the figure caption and added a sideview figure to the Supplementary material.**

Also, is it a coincidence that the more highly fractured zones at the top and bottom of the borehole correspond to the zones of higher uncertainty and also the zones where your array coverage is poorest (i.e. larger azimuthal gaps)? It's a little difficult to make out the network geometry with this one oblique view. Can you provide a map-view and a cross section along with one of the figures (e.g. Figure 1, 4a, or 5)? **Velocity model misfit (isotropic velocity model performing better in fractured zones, see updated Figure 4b) and network geometry play a role in determining the length and the direction of the black uncertainty estimates in Figure 4a. Azimuthal gaps are comparable for the vertical validation borehole and the sections in the injection and long inclined validation borehole, where the location uncertainty is largest.**

Figure 4a: All of the larger uncertainties point upwards. Is this a product of such a one-sided array with only 1 sensor (AE hydrophone) below the stimulation intervals? Or, as you state, an underestimation of the attenuation in the fractures zones? **This is predominantly a network geometry effect (as stated above). During the follow-on project STIMTEC-X, which uses more hydrophones near the fracture zones to obtain a better 3-D coverage, which prevented this one-sided elongation.**

Ln 549: What did this reveal about the detection limits? **By detecting lots of events from shallower depth this posed the question of why no events were detected for deeper stimulation intervals. The active UT measurements (which have comparable amplitude to the AE events, but higher frequency content) showed no detection limitations for the deeper sections of the borehole (obtained to 56 m borehole depth similar to stimulations). For this reason, we installed the hydrophone which has reduced detection limits compared to an AE sensor but in retrospect we have found them to be <17 m. We added information on distance between hydrophone and "aseismic" intervals.**

Ln 569: Can you also show the progressive growth with distance from the injection interval in Figure 6? **We have added this to the figure.**

Ln 571-572: Figure 5 would be a good place for a map-view and cross-section to show the reader that the seismicity from injection and vertical validation are distinct. **We have added two sideview figures in the Supplementary Material to address this.**

Figure 6: A few comments:

• Text is too small in most cases **We enlarged the part where text was too small.**

• AE and Hammer bars in the histogram are indistinguishable **We changed hammer markers to green vertical lines.**

• Very hard to see the waveforms, so what's being shown? You say all events, so are these all the waveforms superimposed on each other, stacked, something else? **The raw recordings of the 7 AE events observed during the stimulation of this interval are shown as recorded by one of the AE sensors. We clarified this in the figure caption.**

• What about the other seismically-active intervals? **These will be shown in our next paper.**

Ln 586: It's hard to assess the claim of that these signals resemble pressure and not flow without adding the hydraulic parameters to these plots in some way. **This statement was based on a visual comparison of the filtered signals. In response to this comment, we have calculated the cross-correlation between pressure and broadband signals and flow and broadband signals, respectively. We have checked filtered versus unfiltered as well as smoothed versus original but resampled data and we observed that the correlation with flow is systematically larger (except for two intervals). We have therefore revised this statement accordingly. We added a table to the Supplementary Material showing these cross-correlation values to substantiate this analysis. We also added a figure to the Supplementary Material showing a comparison of the resampled and smoothed broadband sensor data, pressure and flow data for one interval.**

Figure 7: The discussion of the broadband sensors seems a little tangential to the rest of the paper which I see as focused on the AE locations and velocity model. It might be better to leave this out or transfer to the supplements? **We have left this paragraph, because this manuscript is a description and evaluation of the monitoring system. We have made the suggested changes in related comment l.648.**

Ln 601: Plus no (good) S-wave. **Not changed, the analysis of S-waves on AE hydrophones (in the ARMA paper we refer to) is not yet completed, this was a first analysis.**

Ln 607: And destroyed some of the monitoring equipment⋯**Not changed, not really relevant.**

Ln 609-612: It's a difficult trade-off between near-field monitoring and allowing the fracture to grow unimpeded. EGS Collab #2 will have a central injection well with four surrounding open boreholes (all drilled from same wellhead). Seismic instrumentation will be further afield this time in separate grouted wells (in an attempt to avoid what happened last time). Still, we'll likely intercept the fracture with one of the boreholes surrounding injection. As you mention, this significantly affects the fracture's behavior but also allows us to monitor the fracture at multiple locations (e.g. with a straddle packer or 3D displacement probe). But I agree that seismic monitoring should stay out of the way, especially when using high-sensitivity AEs. **Agreed, the EGS Collab experiments seem to be designed from a "commercial perspective" which differs to our research approach.**

Ln 648-652: If you choose to leave the broadband recordings in the main body of the paper, it would be nice to see a comparison of the waveforms and the hydraulic parameters. **Done, see comment l.586.**

Technical corrections: **All adopted as suggested.**

---

## Author Comment (AC2)

We would like to thank Reviewer 2 for his detailed review comments, which we addressed below in **bold**. All changes referred to in the manuscript are marked as tracked changes. We particularly focused on clarifications and changes requested to the figures.

Overall, the manuscript is well-written and clearly articulated. I do share some of the same concerns as reviewer #1, specifically that the manuscript could be refocused to tighten the discussions surrounding the relationship between the ultrasonic transmission (UT), AE locations and hydraulic stimulation.

**We have not changed the structure to refocus the content but we have added more information to the supplementary material on the tangential parts of the manuscript (as Reviewer 1 suggested). Please see the first paragraph in the response to Reviewer 1 for more details.**

Beyond this, I make some minor scientific and technical suggestions below, mostly surrounding clarifications to the figures:

Scientific revisions/clarifications:

1. Line 308-310: Is there a pre-amplification or band-pass stage to your AE data acquisition? **Yes, we added a few sentences about the amplification, filtering and dynamic ranges.**

2. Line 395-397: '…and more emergent, low-signal to noise ratio onsets,….' Does this refer to the s-waves here? Are the s-arrivals being weighted 50% less for relocations? Or do you mean there are two classes of p-waves, sharp and diffuse/emergent? **This refers to P-waves, only, which we now clarified in the text. We intended to downweight emergent picks but after double checking found out that these were given full weight, so revised this statement. Active source S-picks were not included in the velocity model determination or the error assessment as stated in Section 4.1**

3. Line 505-507: What do you mean by the 'best velocity model is tuned to the injection borehole'? This sounds like a sampling bias, because there are more samples here? If so, could you clarify this further? **Yes, there is a sampling bias, because the injection borehole and vertical validation borehole were sounded twice (before and after stimulation and with several orientations), whereas the other boreholes were sounded only once. Nevertheless, this is not unintentional because the velocity model needs to be most accurate at the injection and vertical validation borehole, because this is where most of the AE events occur. It effectively means we have a weight of 4 for data at the injection borehole (because of different orientations at each position), a weight of 2 for data from the vertical validation borehole and weight of 1 elsewhere. To show that the model is still accurate for all boreholes, we changed Figure 4b) to visualise this, see also comment 8.**

4. Lines 631-633: This statement confuses me a little, because in my experience, even aseismic slip has AEs associated with it during lab-scale AE monitoring, arising from grain-grain sliding/fracturing. This goes back to my comment 1 above whether the AEs you're monitoring are predominantly related to the co-seismic stage (likely no pre-amp, so requires more AE intensity and consequently picks fewer events).

   **There is pre-amplification as addressed in 1. We cannot monitor deformation occurring on a smaller scale (e.g. grain scale) than observable by the AE recording system with a frequency band of 1 to 100 k Hz, corresponding to deformation on the cm to dm scale. We follow the definition of Dresen et al 2020, PAGEOPH that aseismic deformation is deformation occurring out of the seismic recording band. Please also note, that during the follow-up experiment STIMTEC-X we re-stimulated and hydraulically tested previously „aseismic" intervals with several AE hydrophones**

placed in close vicinity to the intervals (3–7 m), but did not record any AE activity in the frequency band 1 to 40 kHz, either.

5. Line 718-719: It would be nice to see this correlation associated with pre-existing structures reflected in Figure 8 somehow, potentially by integrating the FMI scans into the figure? **We have marked the sections shown in Fig 2 in Fig 4b and Fig 8.**

6. Lines 720-722: Perhaps I missed this, but how do you estimate velocity and amplitude changes in the UT data? I assume it is some sort of cross-correlation technique, and if so, it would be useful to see the template, i.e., p, s-arrivals, and the amplitudes (peak-to-peak, rms, 0-to-peak or something else). What is the error in these measurements?

   **To determine velocity change, we compared the measured values at 32 m and 33 m depth with the expected value of best fitting anisotropic model for the station at the depths surrounding the significant structure at 32.5 m depth in the injection borehole (see now with uncertainties in Fig 8a). We estimate changes in amplitude for these measurements by determining the difference between the measured value (median absolute amplitude for different window lengths between 0.150 and 0.5 ms) with the expected value for the data point from linear regression of three neighbouring measurements above and below the depth for nearby stations. We added a statement on the uncertainty in the velocity measurements to Section 4.2 and 5.3 as determined from repeated UT measurements from the same points in the borehole. The uncertainty estimates increase slightly for the measurements at 32 m or 33 m depth compared to neighbouring measurement points, but the drop is significant compared to the uncertainty estimates. Estimating the amplitude uncertainty is rather difficult because there are several factors (coupling of UT source and AE sensors, directional amplitude dependence of the UT source and AE sensors) that control repeatability of the measurements, with coupling at the source likely as the largest influencing factor.**

7. All figures of the drftways (eg. 1, 3, 4a etc.) – Are these the same isometric projections? I see the cardinal directions annotated in a couple of them but not all, so it's not quite clear what the orientations of the various drftways, boreholes are. Also, the 5 m scale is not very clear.

   **Yes, that is correct. We added the 3-D coordinate system to the figure and clarified the figure captions.**

8. Figure 3 – I wonder if there's a better way to illustrate the ray-paths because it is not too useful for the lower (deeper?) boreholes since all you see is grey lines.

   **We have added map- and side views for this figure to the Supplementary Material.**

9. Figure 4b – The injection and validation borehole annotations, as well as the pre/post stimulation annotations are unclear relative to the figure and I'm not sure what they refer to.

   **We have changed this figure now showing the location uncertainty estimates obtained for all boreholes (instead of pre- and post-stimulation measurements of the injection and the vertical validation borehole). We extended the caption.**

10. Figure 5 could be more readable with a cross-sectional view in addition to the isometric view. Additionally, I also suggest exploring the possibility of scaling the AE dots by size and/or location uncertainty (depending on which one's more variable).

    **We have added the cross-sectional view to the Supplementary Material figures. We do not have determined AE event size (magnitude), yet. Location uncertainty is largest in the vertical direction with a median value of 3.7 km. It is very similar for the events in the dense cluster and more variable for the more scattered events surrounding it. Displaying it visually only complicated the figure, but we will incorporate this in a future publication that discusses the AE events of individual stimulation intervals in detail.**

Figure 7: I didn't catch the annotations in the figures until my final reading. I would suggest increasing the font size significantly and changing the star color on these.

**We have adopted the proposed changes.**

11. Figure 8: Similar to the vp drop, it seems like there's a recovery at ~45 m. Does this, then, correspond to a less heterogeneous, more competent formation?

    **Yes, that is correct, the abrupt velocity recovery at >47 m is seen outside the damage zone between 42 and 47 m. We have marked the acoustic log sections from Fig.2 to the figure (and added equivalent figures for other boreholes to the Electronic Supplement) to better illustrate the point that prominent structures influence the velocity.**

---

## Referee Report (RR1)

12-20-2021
Chet Hopp
Postdoc
Lawrence Berkeley National Lab

se-2021-84 Reviewer comments:

**Overall assessment:**

The authors have provided a detailed and complete response to my review of "Seismic monitoring of the STIMTEC hydraulic stimulation experiment in anisotropic metamorphic gneiss". I have no further revisions to suggest and congratulate the authors on an enjoyable and informative manuscript.

All the best,

Chet